# A Novel Data-Envelopment Analysis Interval-Valued Fuzzy-Rough-Number Multi-Criteria Decision-Making (DEA-IFRN MCDM) Model for Determining the Efficiency of Road Sections Based on Headway Analysis

Dejan Andjelković [1], Gordan Stojić [2], Nikola Nikolić [3], Dillip Kumar Das [4], Marko Subotić [5] and Željko Stević [6,*]

[1] Faculty of Applied Sciences Niš, University Business Academy in Novi Sad, Dušana Popovića 22a, 18000 Niš, Serbia; dejan.andjelkovic@fpn.rs
[2] Faculty of Technical Sciences, University of Novi Sad, Trg Dositeja Obradovića 6, 21000 Novi Sad, Serbia; gordan@uns.ac.rs
[3] Faculty of Technical Sciences "Mihajlo Pupin" Zrenjanin, University of Novi Sad, Djure Djakovica bb, 23101 Zrenjanin, Serbia; kontakt@nikola-nikolic.com
[4] Sustainable Transportation Research Group, Civil Engineering, School of Engineering, University of Kwazulu Natal, Durban 4041, South Africa; dasd@ukzn.ac.za
[5] Faculty of Transport and Traffic Engineering, University of East Sarajevo, Vojvode Mišića 52, 74000 Doboj, Bosnia and Herzegovina; marko.subotic@sf.ues.rs.ba
[6] College of Engineering, Korea University, 145 Anam-Ro, Seongbuk-gu, Seoul 02841, Republic of Korea
* Correspondence: zeljkostevic88@yahoo.com

**Abstract:** The capacity of transport infrastructure is one of the very important tasks in transport engineering, which depends mostly on the geometric characteristics of road and headway analysis. In this paper, we have considered 14 road sections and determined their efficiency based on headway analysis. We have developed a novel interval fuzzy-rough-number decision-making model consisting of DEA (data envelopment analysis), IFRN SWARA (interval-valued fuzzy-rough-number step-wise weight-assessment-ratio analysis), and IFRN WASPAS (interval-valued fuzzy-rough-number weighted-aggregate sum–product assessment) methods. The main contribution of this study is a new extension of WASPAS method with interval fuzzy rough numbers. Firstly, the DEA model was applied to determine the efficiency of 14 road sections according to seven input–output parameters. Seven out of the fourteen alternatives showed full efficiency and were implemented further in the model. After that, the IFRN SWARA method was used for the calculation of the final weights, while IFRN WASPAS was applied for ranking seven of the road sections. The results show that two sections are very similar and have almost equal efficiency, while the other results are very stable. According to the results obtained, the best-ranked is a measuring segment of the Ivanjska–Šargovac section, with a road gradient = −5.5%, which has low deviating values of headways according to the measurement classes from PC-PC to AT-PC, which shows balanced and continuous traffic flow. Finally, verification tests such as changing the criteria weights, comparative analysis, changing the λ parameter, and reverse rank analysis have been performed.

**Keywords:** road traffic; headway; DEA; IFRN SWARA; IFRN WASPAS

**MSC:** 90B50; 90B20; 90C08

## 1. Introduction

Traffic congestion is one of the leading problems [1] around the world that affects the economy and productivity of countries in the world. Traffic jams are the result of traffic demands at different periods of time. Under congested conditions, drivers usually reduce

the time headway (*Th*), which usually leads to deviations in the stochastic representation of these traffic parameters. This parameter depends not only on traffic conditions, but also on the drivers' behavior during different traffic scenarios. The headway of vehicles represents an important microscopic traffic-flow parameter that can be defined as the time difference between the passage of the fronts of any two consecutive vehicles on an observed road section [2]. This parameter is used in various research areas applied to traffic, starting with road capacity, road safety, traffic efficiency, as well as many other indicators based on stochastic traffic phenomena. That is why modeling and an analytical description of headway play a key role in obtaining traffic analyses and making relevant decisions related to road infrastructure. A large number of studies applied different probability models, machine learning algorithms, as well as neural networks for the short-term prediction of headways. Nevertheless, headway is characterized as a random variable, varying from case to case, and is especially functionally dependent on road and traffic characteristics, as well as characteristics of the environment, climate, altitude, and other influences.

Headway as a traffic-flow parameter particularly affects traffic safety, the level of service, and road capacity. This parameter, when observing the following between two vehicles, depends on the type of the leading vehicle and on its vehicle dynamic characteristics. The influence of the leading vehicle is especially evident on two-lane two-way roads with a heterogeneous flow structure where there are a large number of different types of vehicles.

The contributions and novelty of the paper are reflected in the following facts:

- In accordance with the previously defined importance of headways for the entire area of road transport, a total of 14 sections of road infrastructure were considered and a new model to determine their efficiency was created. In addition to five headway classifications, AADT (annual average daily traffic) and road gradient were taken as influential parameters.
- A new multiphase efficiency model, which includes the DEA model, SWARA, and WASPAS methods in the form of interval fuzzy rough numbers, was created. IFRNs were used due to their ability to treat uncertainty in the decision-making process adequately. The greatest contribution of the paper can be seen from the aspect of a new algorithm of the IFRN WASPAS method, which, according to the authors' knowledge, is presented for the first time in the literature. So far, certain comparative analyses have been presented, but without the algorithm of this method.
- Another aspect of the contribution is reflected through the sustainable management of road infrastructure based on the results obtained and future recommendations. From a practical aspect, the study provides valuable insights for infrastructure managers and traffic experts, helping them make informed decisions to optimize road section efficiency.

Research gaps are described in the following sections. The paper thoroughly analyzes road-section efficiency by considering multiple input–output parameters, determining criteria weights by applying the IFRN SWARA method, sorting road sections with the IFRN WASPAS method, and performing verification tests. This comprehensive model enhances the credibility and reliability of the research findings.

Further in the paper, Section 2 provides a review of the literature that considers headway as a basis for defining input parameters. In Section 3, an overall procedure using the applied methods is presented with an emphasis on a new algorithm of the IFRN WASPAS method. Section 4 presents the formulation of the multiphase model with subsections that refer individually to the application of each method. It is important to emphasize that the procedure for applying the model is explained in detail by phases. Section 5 includes verification tests through four extensive analyses, while the paper ends with a conclusion in Section 6.

## 2. Literature Review

Vehicles that move in conditions of free traffic flow are vehicles that move individually or follow other vehicles. When the traffic flow reaches the saturation value, headway tends

to the minimum value. Based on these traffic conditions, a large number of researchers analyzed headway with the aim of obtaining representative values. Based on research conducted in Pakistan [3] on two-lane roads with heterogeneous traffic, which included traffic flow, headway, and traffic density, the moving method (MM) was compared with the most commonly used stationary method (SM). A linear model was used to obtain headways by applying the moving method. Also, in the research on two-lane two-way roads in northern Italy [4], a set of headway distribution models was tested by the statistical analysis of data obtained from radar sensors and inductive loops. Research on four measurement sections has shown that an inverse Weibull distribution is the most suitable for representing headways for most flow-rate ranges. A study conducted in Iran [5] on the Shahid Kharrazi six-lane highway investigated the influence of lane position on the time-headway distribution under a high level of traffic flow. The appropriate model of headway distribution is based on the $\chi^2$ test for each traffic lane, where the results of the study confirm the assumption that the appropriate model for the passing lane is different from the model for the middle lane. By following cars in the passing lane, a large number of drivers adopt unsafe headways leading to significant differences in the capacity and statistical models of headway distribution for different lanes. Also, a significant number of studies [3,6–8], in addition to headway, use other traffic-flow parameters (free speed, density, flow rate, etc.) to assess traffic conditions on road sections, where the 85th percentile values are often analyzed. In the research described in [6], it was concluded that headway cannot be used to identify free-flow speed on multi-lane roads, because this interval depends on the length of vehicles. Also, in this research, credibility values of speeds (−7 km/h to +15 km/h) were analyzed, where a gap of 10 s was identified, and this was used to identify the next vehicle on four highway lanes under heterogeneous traffic conditions. Also, research in India [9] on two-lane roads showed headway values distributed according to the log-logistic distribution in conditions of moderate flow-rate values, and Pearson-5 distribution was used in conditions of congestion. This was selected out of four considered distributions.

Veng et al. [10] proved that the type of vehicle has a significant influence on headway distribution, and the scenarios of car–car, car–truck, truck–truck and truck–car rarely appear in real traffic conditions, so the headway distribution model is analyzed separately for different types of vehicles. Also, by analyzing headway in the conditions of heterogeneous traffic flow, the truck–car ratio is included as one of specific variables for determining functional parameters [11]. Based on the research carried out in Iraq [12] on over 8000 headways, in order to determine the critical headway, it was found that the range of the critical interval was from 2.5 to 4 s, with a corresponding critical headway of 3.2 s on 10 measuring sections of the two-lane highways. The best model for headway distribution in free-flow conditions is with a negative exponential distribution, while in conditions with vehicle restrictions, it is a lognormal distribution model. By analyzing seven probability headway distributions on two rural two-lane two-way roads in Egypt [13], one-hour videotaped data were collected, and they showed that gamma and shifted exponential distributions are appropriate distributions for modeling headways in the Dakahliya province.

The headway research [14] was conducted by considering the influence of lateral distances between vehicles moving on the roadway in different lanes. In this study, driving behavior, speed/headway relationship, and the following threshold were investigated, with headways being segmented into five classes: unsafe (0–0.7 s), non-lane-based car following (0.9 s), lane-based car following (1.0 s), overtaking (1.3 s) and free driving (over 2.5 s). A linear relationship was found between time headway and lateral distances in non- lane-based car-following conditions. Also, when observing the behavior of vehicles in two different lanes, an insignificant lateral distance between the following and preceding vehicles was shown for the lowest headway value.

Research [15] based on deep learning showed that there is no suitable model for the long-term prediction of traffic headways, since current models do not use a large data set and do not solve the problem of a longitudinal gradient. The obtained headway values

are not of constant size in the same ambient conditions, so they mainly depend on the drivers' perceptive ability, processing of the received data, actions taken, and heterogeneous vehicle performance [16]. In the research [11] based on the use of two sets of experimental data for the calculation of headway values, 18 commonly used value distribution models were applied in order to select the best model. The study demonstrated a distribution model with adaptive parameters, and its performance and applicability were verified. The performance of the model was improved by 62.7% compared with the model with fixed parameters. Also, on the basis of sixteen pairs of vehicles identified in the field, the movement of specific types of vehicles in a heterogeneous flow on a national road in Northeast India was analyzed [17]. The observations in that study showed that car drivers have a conservative attitude and usually keep a safe distance from the leading vehicle. In addition, a comparison of computed headway probabilities was made with the values obtained from more or less homogeneous traffic. The values obtained in the current study were found to be high in most cases, indicating risk-taking driver behavior. Also, to determine headway in the study [4] on Italian two-lane roads, an exponential moving model was introduced in order to identify a criterion above which vehicle movements could be considered unconditional. However, by applying this model, it was possible to identify vehicles that still have a certain autonomy in their speed and maneuvering, so an additional criterion was introduced to distinguish apparently and truly conditioned vehicles by analyzing the differences in the speeds of vehicles following each other.

In order to improve the quality of two-lane roads and model prediction, the effects of vehicle driving variables on road-performance measures were evaluated, and then critical headways were identified with the aim of accepting optimal gaps between vehicles. In order to achieve the research objective, multiple linear-regression and Bayesian linear-regression models were developed, which showed a headway threshold of 2.4 s based on vehicular platooning [18]. Also, a platoon is defined as a series of vehicles where the time interval between the leading and following vehicles is 3 s [19,20]. Additionally, it should be noted that in the past, the time headway limit was 5 s [21], while more recent recommendations indicate a value of 3 s [19,22]. The research conducted on 50 different sections of rural two-lane roads classified into two classes in Serbia shows that there is a difference in headway-limit values for two classes of roads in free-flow conditions. For class I, headway was 6.3 s, and for class II, it was 8.4 s [23]. A capacity survey conducted on the Benin—Lagos road section in Edo State, Nigeria, showed an average space headway of 0.025 km (25 m) and an average time headway of 2.26 s, indicating a moderate traffic flow. The values obtained in this way show a low probability of traffic accidents [7].

A study based on determining time gaps between vehicles (the rear part of a vehicle and the front part of its follower) was conducted on 13 km of the airport access road in Washington, using a sample of 168,053 time gaps. The study showed significant variations in the values of the time gaps. Also, at speeds above 108 km/h, the minimum time gaps made by some drivers could be 1.6 times longer compared with the minimum values made by other drivers [24].

## 3. Methods

### 3.1. Preliminaries

This approach provides three rough sets expressed as an interval. The expression obtained is called the interval fuzzy rough number "$A$".

$$A = \left[ A_q^L, A_q^u \right] = \left[ \left( a_{1q}^L, a_{1q}^U \right), \left( a_{2q}^L, a_{2q}^U \right), \left( a_{3q}^L, a_{3q}^U \right) \right] \tag{1}$$

where $a_{jq}^L = \underline{Lim} \left( I * \left( a_j \right)_{lq} \right)$ and $a_{jq}^U = \overline{Lim} \left( I * \left( a_j \right)_{uq} \right); (j = 1, 2, 3; 1 \leq q \leq k)$

The interval fuzzy rough number $A$ defined in the interval $(-\infty, +\infty)$ can be shown by Equations (2) and (3) [25,26].

$$A = \left\{ x, \left[ \mu_{A_q^L}(x), \mu_{A_q^U}(x) \right] \right\}, x \in (-\infty, +\infty), \mu_{A_q^L}(x), \mu_{A_q^U}(x) : (-\infty, +\infty) \rightarrow [0,1] \quad (2)$$

$$\mu_A(x) = \left[ \mu_{A_q^L}(x), \mu_{A_q^U}(x) \right], \mu_{A_q^L}(x) \leq \mu_{A_q^U}(x), \forall x \in (-\infty, +\infty) \quad (3)$$

The values $\mu_{A_q^L}(x)$ and $\mu_{A_q^U}(x)$ are the degree of membership to the lower and upper functions, respectively, of the interval fuzzy rough number $A$.

When manipulating with two interval fuzzy rough numbers $A$ and $B$, different mathematical operations between them can be performed.

Addition of two interval fuzzy rough numbers:

$$\begin{aligned} A + B &= \left[ (a_1^L, a_1^U), (a_2^L, a_2^U), (a_3^L, a_3^U) \right] + \left[ (b_1^L, b_1^U), (b_2^L, b_2^U), (b_3^L, b_3^U) \right] \\ &= \left[ (a_1^L + b_1^L, a_1^U + b_1^U), (a_2^L + b_2^L, a_2^U + b_2^U), (a_3^L + b_3^L, a_3^U + b_3^U) \right] \end{aligned} \quad (4)$$

Subtraction of two interval fuzzy rough numbers:

$$\begin{aligned} A - B &= \left[ (a_1^L, a_1^U), (a_2^L, a_2^U), (a_3^L, a_3^U) \right] - \left[ (b_1^L, b_1^U), (b_2^L, b_2^U), (b_3^L, b_3^U) \right] \\ &= \left[ (a_1^L - b_3^U, a_1^U - b_1^L), (a_2^L - b_2^U, a_2^U - b_2^L), (a_3^L - b_1^U, a_3^U - b_1^L) \right] \end{aligned} \quad (5)$$

Multiplication of two interval fuzzy rough numbers:

$$\begin{aligned} A \times B &= \left[ (a_1^L, a_1^U), (a_2^L, a_2^U), (a_3^L, a_3^U) \right] \times \left[ (b_1^L, b_1^U), (b_2^L, b_2^U), (b_3^L, b_3^U) \right] \\ &= \left[ (a_1^L \times b_1^L, a_1^U \times b_1^U), (a_2^L \times b_2^L, a_2^U \times b_2^U), (a_3^L \times b_3^L, a_3^U \times b_3^U) \right] \end{aligned} \quad (6)$$

Division of two interval fuzzy rough numbers:

$$\begin{aligned} A \div B &= \left[ (a_1^L, a_1^U), (a_2^L, a_2^U), (a_3^L, a_3^U) \right] \div \left[ (b_1^L, b_1^U), (b_2^L, b_2^U), (b_3^L, b_3^U) \right] \\ &= \left[ (a_1^L \div b_3^U, a_1^U \div b_1^L), (a_2^L \div b_2^U, a_2^U \div b_2^L), (a_3^L \div b_1^U, a_3^U \div b_1^L) \right] \end{aligned} \quad (7)$$

### 3.2. Interval Fuzzy-Rough-Number SWARA Method

We have used this subjective method for calculating criteria weights because this case study needs the adequate expertise of DMs and their preferences based on skills and knowledge. This method is extended with numerous theories [27–30] and, finally, with the IFRN [31].

Step 1: Formation of a group of $m$ criteria.

Step 2: Definition of a team of $e$ experts to evaluate the criteria. Experts can use any of the given scales to determine the significance of the criteria.

Step 3: Transformation of individual experts' estimates into a group fuzzy rough matrix $x_j$.

$$IFRN(X_j) = \left[ \left( x_j^{L1}, x_j^{U1} \right), \left( x_j^{L2}, x_j^{U2} \right), \left( x_j^{L3}, x_j^{U3} \right) \right]_{1 \times n} \quad (8)$$

Step 4. Ranking of criteria by their significance obtained using the fuzzy rough matrix from Step 3.

Step 5: Normalization of the matrix $IFRN(X_j)$ in order to gain the matrix $IFRN(N_j)$

$$IFRN(N_j) = \left[ \left( n_j^{L1}, n_j^{U1} \right), \left( n_j^{L2}, n_j^{U2} \right), \left( n_j^{L3}, n_j^{U3} \right) \right]_{1 \times n} \quad (9)$$

The elements of the matrix $IFRN(N_j)$ are calculated as follows:

$$IFRN(N_j) = \frac{IFRN(X_j)}{IFRN(Z_j)} \quad (10)$$

where $IFRN(Z_j) = \left[\left(z_j^{L1}, z_j^{U1}\right), \left(z_j^{L2}, z_j^{U2}\right), \left(z_j^{L3}, z_j^{U3}\right)\right] = \max IFRN(X_j)$.

The first element of $IFRN(N_j)$, i.e.,

$\left[\left(n_j^{L1}, n_j^{U1}\right), \left(n_j^{L2}, n_j^{U2}\right), \left(n_j^{L3}, n_j^{U3}\right)\right] = [(1.00, 1.00), (1.00, 1.00), (1.00, 1.00)]$, because $j = 1$. For other elements where $j > 1$, Equation (11) should be applied:

$$IFRN(N_j) = \left[\left(\frac{n_j^{L1}}{z_j^{U3}}, \frac{n_j^{U1}}{z_j^{L3}}\right), \left(\frac{n_j^{L2}}{z_j^{U2}}, \frac{n_j^{U2}}{z_j^{L2}}\right), \left(\frac{n_j^{L3}}{z_j^{U1}}, \frac{n_j^{U3}}{z_j^{L1}}\right)\right]_{1\times n} \quad j = 2, 3, ..., n \qquad (11)$$

In the case where there are two most significant criteria, the second element is a fuzzy rough number $[(1.00, 1.00), (1.00, 1.00), (1.00, 1.00)]$.

Step 6: Computation of the matrix $IFRN(\Im_j)$:

$$IFRN(\Im_j) = \left[\left(\Im_j^{L1}, \Im_j^{U1}\right), \left(\Im_j^{L2}, \Im_j^{U2}\right), \left(\Im_j^{L3}, \Im_j^{U3}\right)\right]_{1\times n} \qquad (12)$$

by Equation (13):

$$IFRN(\Im_j) = \left[\left(n_j^{L1} + 1, n_j^{U1} + 1\right), \left(n_j^{L2} + 1, n_j^{U2} + 1\right), \left(n_j^{L3} + 1, n_j^{U3} + 1\right)\right]_{1\times n} j = 2, 3, ..., n \qquad (13)$$

In the case where there are two most significant criteria, the second element is a fuzzy rough number $[(1.00, 1.00), (1.00, 1.00), (1.00, 1.00)]$.

Step 7: Computation of the matrix of recalculated weights $IFRN(\Re_j)$:

$$IFRN(\Re_j) = \left[\left(\Re_j^{L1}, \Re_j^{U1}\right), \left(\Re_j^{L2}, \Re_j^{U2}\right), \left(\Re_j^{L3}, \Re_j^{U3}\right)\right]_{1\times n} \qquad (14)$$

The elements of the matrix $IFRN(\Re_j)$ are obtained as follows:

$$IFRN(\Re_j) \left[ \begin{array}{cc} \Re_j^{L1} = \left(\begin{array}{c} 1.00 \quad j = 1 \\ \frac{\Re_{j-1}^{L1}}{\Im_j^{U3}} \quad j > 1 \end{array}\right), & \Re_j^{U1} = \left(\begin{array}{c} 1.00 \quad j = 1 \\ \frac{\Re_{j-1}^{U1}}{\Im_j^{L3}} \quad j > 1 \end{array}\right), \\ \Re_j^{L2} = \left(\begin{array}{c} 1.00 \quad j = 1 \\ \frac{\Re_{j-1}^{L2}}{\Im_j^{U2}} \quad j > 1 \end{array}\right), & \Re_j^{U2} = \left(\begin{array}{c} 1.00 \quad j = 1 \\ \frac{\Re_{j-1}^{U2}}{\Im_j^{L2}} \quad j > 1 \end{array}\right), \\ \Re_j^{L3} = \left(\begin{array}{c} 1.00 \quad j = 1 \\ \frac{\Re_{j-1}^{L3}}{\Im_j^{U1}} \quad j > 1 \end{array}\right), & \Re_j^{U3} = \left(\begin{array}{c} 1.00 \quad j = 1 \\ \frac{\Re_{j-1}^{U3}}{\Im_j^{L1}} \quad j > 1 \end{array}\right), \end{array} \right] \qquad (15)$$

In the case where any two of $n$ criteria have equal importance, then the following equation should be used:

$$IFRN(\Re_j) = IFRN(\Re_{j-1}) \qquad (16)$$

Step 8: Calculation of final weight values $IFRN(W_j)$:

$$IFRN(W_j) = \left[\left(w_j^{L1}, w_j^{U1}\right), \left(w_j^{L2}, w_j^{U2}\right), \left(w_j^{L3}, w_j^{U3}\right)\right]_{1\times n} \qquad (17)$$

Individual weight values of the criteria are obtained as follows:

$$IFRN(W_j) = \left[\frac{IFRN(\Re_j)}{IFRN(\aleph_j)}\right] \qquad (18)$$

where $IFRN(\aleph_j) = \sum\limits_{j=1}^{n} IFRN(\Re_j)$. Finally,

$$IFRN(W_j) = \left[ \left( \frac{\Re_j^{L1}}{\aleph_j^{U3}}, \frac{\Re_j^{U1}}{\aleph_j^{L3}} \right), \left( \frac{\Re_j^{L2}}{\aleph_j^{U2}}, \frac{\Re_j^{U2}}{\aleph_j^{L2}} \right), \left( \frac{\Re_j^{L3}}{\aleph_j^{U1}}, \frac{\Re_j^{U3}}{\aleph_j^{L1}} \right) \right]_{1 \times n} \quad j = 2, 3, ..., n \quad (19)$$

### 3.3. A Novel Interval Fuzzy-Rough-Number WASPAS Method

This section of the paper is devoted to the development of the interval fuzzy-rough-number WASPAS method, which, in its various forms [32,33], was applied to solve different problem structures. A new algorithm, which includes the extension of the WASPAS method with the IFRN, is presented below.

Step 1. Form a set of alternatives and influential criteria.

Step 2. Since group decision-making is assumed, it is necessary to define a set of DMs that will evaluate potential alternatives.

Step 3. Transform linguistic variables into interval fuzzy rough numbers and form an initial decision matrix as shown:

$$IFRN(A_{ij}) = \begin{bmatrix} (a_{11}^{L1}, a_{11}^{U1}), (a_{11}^{L2}, a_{11}^{U2}), (a_{11}^{L3}, a_{11}^{U3}) & \cdots & (a_{1n}^{L1}, a_{1n}^{U1}), (a_{1n}^{L2}, a_{1n}^{U2}), (a_{1n}^{L3}, a_{1n}^{U3}) \\ (a_{21}^{L1}, a_{21}^{U1}), (a_{21}^{L2}, a_{21}^{U2}), (a_{21}^{L3}, a_{21}^{U3}) & \cdots & (a_{2n}^{L1}, a_{2n}^{U1}), (a_{2n}^{L2}, a_{2n}^{U2}), (a_{2n}^{L3}, a_{2n}^{U3}) \\ \cdots & \vdots & \cdots \\ (a_{m1}^{L1}, a_{m1}^{U1}), (a_{m1}^{L2}, a_{m1}^{U2}), (a_{m1}^{L3}, a_{m1}^{U3}) & \cdots & (a_{mn}^{L1}, a_{mn}^{U1}), (a_{mn}^{L2}, a_{mn}^{U2}), (a_{mn}^{L3}, a_{mn}^{U3}) \end{bmatrix} \quad (20)$$

Step 4: Determine the normalized values that make up the matrix $IFRN(D_{ij})$, which is obtained as follows:

$$d_{ij} = \left[ \left( \frac{a_{ij}^{L1}}{\max a_{ij}^{U3}}, \frac{a_{ij}^{U1}}{\max a_{ij}^{L3}} \right), \left( \frac{a_{ij}^{L2}}{\max a_{ij}^{U2}}, \frac{a_{ij}^{U2}}{\max a_{ij}^{L2}} \right), \left( \frac{a_{ij}^{L3}}{\max a_{ij}^{U1}}, \frac{a_{ij}^{U3}}{\max a_{ij}^{L1}} \right) \right] \; for \; B \quad (21)$$

$$d_{ij} = \left[ \left( \frac{\min a_{ij}^{L1}}{a_{ij}^{U3}}, \frac{\min a_{ij}^{U1}}{a_{ij}^{L3}} \right), \left( \frac{\min a_{ij}^{L2}}{a_{ij}^{U2}}, \frac{\min a_{ij}^{U2}}{a_{ij}^{L2}} \right), \left( \frac{\min a_{ij}^{L3}}{a_{ij}^{U1}}, \frac{\min a_{ij}^{U3}}{a_{ij}^{L1}} \right) \right] \; for \; C \quad (22)$$

Step 5. Integrate normalized matrix values $IFRN(D_{ij})$ with criteria weights $IFRN(W_j)$.

$$v_{ij} = \left[ \left( a_{ij}^{L1} \times w_j^{L1}, a_{ij}^{U1} \times w_j^{U1} \right), \left( a_{ij}^{L2} \times w_j^{L2}, a_{ij}^{U2} \times w_j^{U2} \right), \left( a_{ij}^{L3} \times w_j^{L3}, a_{ij}^{U3} \times w_j^{U3} \right) \right] \quad (23)$$

Step 6. Summarize the IFRN values by rows in order to determine the sum-weighted model $IFRN(S_{ij})$.

$$IFRN(S_{ij}) = \sum_{j=1}^{n} IFRN(v_{ij}) \quad (24)$$

Step 7. Determine the product-weighted function $IFRN(Q_{ij})$ as follows:

$$IFRN(Q_{ij}) = \prod_{j=1}^{n} (d_{ij})^{w_j} \quad (25)$$

Step 8. Rank the alternatives in descending order based on the calculated final values:

$$IFRN(T_{ij}) = \lambda \times IFRN(S_{ij}) + (1 - \lambda) \times IFRN(Q_{ij}), \; \lambda = 0 - 1 \quad (26)$$

## 4. Efficiency of Road Sections Based on Headway Analysis

### 4.1. Collection and Processing of Data

In order to determine the road-section efficiency based on headways, data was collected for a total of 14 sections with the following characteristics: The length of a measuring segment along which the measurements were made is at least 1000 m long, the road gradient ranges from −5.50 to 7.50%, the section length is from 7.45 km to 38.55 km, while the size of the measurement sample varies from 713 to 1011. Also, it is important to note that headway was measured for all types of vehicles. The characteristics of the measuring sections are presented in Table 1.

**Table 1.** Characteristics of road sections and average headways.

| | | Technical and Operational Characteristics of the Sections | | | Measurement Sample Size (No. of Measurements) | AM of Headways by Measuring Segments of Sections | | | | |
|---|---|---|---|---|---|---|---|---|---|---|
| | Section Symbol | Measuring-Segment Length | Road Gradient | Section Length (km) | | Th [s] (PC-PC) | Th [s] (LDV-PC) | Th [s] (HDV-PC) | Th [s] (BUS-PC) | Th [s] (AT-PC) |
| DMU1 | M-I-108 | min 1000 m | −5.50% | 14.967 | 1000 | 5.349 | 7.714 | 7.577 | 8.633 | 10.136 |
| DMU2 | M-I-103 | min 1000 m | −5.00% | 14.073 | 1000 | 19.269 | 25.272 | 40.269 | 29.491 | 44.767 |
| DMU3 | M-I-108 | min 1000 m | −3.00% | 14.967 | 1000 | 5.331 | 9.406 | 9.107 | 4.979 | 7.934 |
| DMU4 | M-I-108 | min 1000 m | 1.50% | 14.967 | 1000 | 12.553 | 21.713 | 19.393 | 23.681 | 12.926 |
| DMU5 | M-I-103 | min 1000 m | −1.00% | 14.073 | 1010 | 18.174 | 21.102 | 36.395 | 26.04 | 33.222 |
| DMU6 | M-I-105 | min 1000 m | 0% | 7.405 | 1011 | 3.6 | 8.609 | 11.969 | 10.162 | 16.488 |
| DMU7 | M-I-106 | min 1000 m | 1.00% | 38.553 | 912 | 9.794 | 25.096 | 18.537 | 26.949 | 16.669 |
| DMU8 | M-I-108 | min 1000 m | 2.00% | 16.734 | 775 | 24.22 | 65.23 | 66.46 | 64.88 | 97.99 |
| DMU9 | M-I-106 | min 1000 m | 3.00% | 38.553 | 908 | 12.43 | 31.124 | 24.458 | 21.191 | 64.944 |
| DMU10 | M-I-106 | min 1000 m | 4.00% | 38.553 | 1007 | 8.48 | 31.271 | 31.253 | 34.794 | 63.897 |
| DMU11 | M-I-103 | min 1000 m | 5.00% | 14.073 | 918 | 4.79 | 32.367 | 36.991 | 36.362 | 82.858 |
| DMU12 | M-I-108 | min 1000 m | 6.00% | 20.134 | 736 | 12.907 | 82.888 | 126.711 | 118.67 | 189.593 |
| DMU13 | M-I-108 | min 1000 m | 7.00% | 20.134 | 713 | 14.739 | 90.027 | 132.791 | 135.949 | 236.574 |
| DMU14 | M-I-108 | min 1000 m | 7.50% | 20.134 | 811 | 16.559 | 97.84 | 141.196 | 146.706 | 264.434 |

DMU (Decision-making unit); PC (Passenger car); LDV (Light-Duty Vehicle); HDV (Heavy-Duty Vehicle); AT (Auto train).

After the collection, processing, and sorting of data, a DEA model [34] was applied based on a total of seven parameters, which later represent criteria in the MCDM model. In addition to the given parameters related to the headway and road gradient, AADT was in-cluded as an additional parameter. *C1*—AADT; *C2*—road gradient; *C3*—Th [s] (PC-PC); *C4*—Th [s] (LDV-PC); *C5*—Th [s] (HDV-PC); *C6*—Th [s] (BUS-PC); *C7*—Th [s] (AT-PC).

### 4.2. Application of DEA Model

The results after applying the DEA model are as follows:

$$DMU1 = 1.000, \ DMU2 = 0.119, \ DMU3 = 1.000, \ DMU4 = 1.000, \ DMU5 = 0.153, \ DMU6 = 1.000, \ DMU7 = 0.667$$
$$DMU8 = 0.315, \ DMU9 = 1.000, \ DM10 = 1.000, \ DMU11 = 1.000, \ DMU12 = 0.519, \ DMU13 = 0.503, \ DMU14 = 0.496$$

This means that half of the road sections are efficient in terms of the observed parameters, namely DMU1, DMU3, DMU4, DMU6, DMU9, DMU10, and DMU11. Since the discriminatory power in the DEA model is only 50% in this case, the IFRN MCDM model is defined in order to finally determine the efficiency for each road section.

### 4.3. Determining the Importance of Parameters Using the IFRN SWARA Method

In this section of the paper, it is first necessary to define the mutual relationship between the criteria, which was carried out by three decision-makers (DMs), and this is shown in Table 2.

**Table 2.** DM assessment of criteria significance.

|  | Criterion | DM1 | DM2 | DM3 |
|---|---|---|---|---|
| C1 | AADT | (3,4,5) | (4,5,6) | (4,5,6) |
| C2 | Road gradient | (4,5,6) | (4,5,6) | (5,6,7) |
| C3 | Th [s] (PC-PC) | (3,4,5) | (3,4,5) | (3,4,5) |
| C4 | Th [s] (LDV-PC) | (2,3,4) | (3,4,5) | (3,4,5) |
| C5 | Th [s] (HDV-PC) | (2,3,4) | (2,3,4) | (1,2,3) |
| C6 | Th [s] (BUS-PC) | (1,2,3) | (2,3,4) | (0,1,2) |
| C7 | Th [s] (AT-PC) | (0,1,2) | (1,2,3) | (1,2,3) |

In order to be able to apply the steps of the IFRN SWARA method, it is necessary to convert the DMs' estimates into interval fuzzy rough numbers.

A rough matrix for *C7* is obtained as follows:

According to the DMs' estimates given in Table 2, three classes of objects are selected, i.e., *l*, *m*, and *u*, where *l* = (0;1;1), *m* = (1;2;2), and *u* = (2;3;3).

For *l*:

$$\underline{Lim}(0) = 0, \overline{Lim}(0) = \frac{1}{3}(0 + 1 + 1) = 0.667; \ \underline{Lim}(1) = \frac{1}{3}(0 + 1 + 1) = 0.667, \overline{Lim}(1) = 1$$

For *m*:

$$\underline{Lim}(1) = 1, \overline{Lim}(1) = \frac{1}{3}(1 + 2 + 2) = 1.667; \ \underline{Lim}(2) = \frac{1}{3}(1 + 2 + 2) = 1.667, \overline{Lim}(2) = 2$$

For *u*:

$$\underline{Lim}(2) = 2, \overline{Lim}(2) = \frac{1}{3}(2 + 3 + 3) = 2.667; \underline{Lim}(3) = \frac{1}{3}(2 + 3 + 3) = 2.667, \overline{Lim}(3) = 3$$

Thus, *IFRNs* are obtained as follows:

$$IFRN(DM_1) = [(0.00, 0.67), (1.00, 1.67), (2.00, 2.67)]$$

$$IFRN(DM_2) = [(0.67, 1.00), (1.67, 2.00), (2.67, 3.00)]$$

$$IFRN(DM_3) = [(0.67, 1.00), (1.67, 2.00), (2.67, 3.00)].$$

Using an aggregation equation, the final fuzzy rough number for *C7* is computed:

$$IFRN(C_7) = [(0.45, 0.89), (1.45, 1.89), (2.45, 2.89)]$$

and the final interval fuzzy rough matrix $IFRN(X_j)$ is obtained (Table 3).

**Table 3.** Initial fuzzy rough matrix in the IFRN SWARA method.

|  | *Xj* |
|---|---|
| C7 | [(0.447,0.890),(1.447,1.890),(2.447,2.890)] |
| C6 | [(0.500,1.500),(1.500,2.500),(2.500,3.500)] |
| C5 | [(1.447,1.890),(2.447,2.890),(3.447,3.890)] |
| C4 | [(2.447,2.890),(3.447,3.890),(4.447,4.890)] |
| C3 | [(3.000,3.000),(4.000,4.000),(5.000,5.000)] |
| C1 | [(3.447,3.890),(4.447,4.890),(5.447,5.890)] |
| C2 | [(4.110,4.553),(5.110,5.553),(6.110,6.553)] |

The normalized matrix $IFRN(X_j)$ given below is obtained in the following way:
The first element of the matrix $IFRN(N_j)$, i.e.,

$$[(n_7^{L1}, n_7^{U1}), (n_7^{L2}, n_7^{U2}), (n_7^{L3}, n_7^{U3})] = [(1.000, 1.000), (1.000, 1.000), (1.000, 1.000)],$$ represents a rule, and it is required in each decision-making process.

$$IFRN(N_j) = \begin{bmatrix} (1.000, 1.000), (1.000, 1.000), (1.000, 1.000) \\ (0.076, 0.245), (0.270, 0.489), (0.549, 0.852) \\ (0.221, 0.309), (0.441, 0.556), (0.757, 0.946) \\ (0.373, 0.473), (0.621, 0.761), (0.977, 1.190) \\ (0.458, 0.491), (0.720, 0.783), (1.098, 1.217) \\ (0.526, 0.637), (0.801, 0.957), (1.196, 1.433) \\ (0.627, 0.745), (0.920, 1.087), (1.342, 1.594) \end{bmatrix}$$

$$IFRN(Z_j) = [(4.11, 4.55), (5.11, 5.55), (6.11, 6.55)]$$

$$[(n_6^{L1}, n_6^{U1}), (n_6^{L2}, n_6^{U2}), (n_6^{L3}, n_6^{U3})] = \left[\left(\frac{0.50}{6.55}, \frac{1.50}{6.11}\right), \left(\frac{1.50}{5.55}, \frac{2.50}{5.11}\right), \left(\frac{2.50}{4.55}, \frac{3.50}{4.11}\right)\right] = [(0.076, 0.245), (0.270, 0.489), (0.549, 0.852)]$$

The next step is to compute the following interval fuzzy rough matrix:

$$IFRN(\Im_6) = [(0.076 + 1.00, 0.245 + 1), (0.270 + 1.00, 0.489 + 1.00), (0.549 + 1.00, 0.852 + 1.00)] = [(1.076, 1.245), (1.270, 1.489), (1.549, 1.852)]$$

$$IFRN(\Im_j) = \begin{bmatrix} (1.000, 1.000), (1.000, 1.000), (1.000, 1.000) \\ (1.076, 1.245), (1.270, 1.489), (1.549, 1.852) \\ (1.221, 1.309), (1.441, 1.566), (1.757, 1.946) \\ (1.373, 1.473), (1.621, 1.761), (1.977, 2.190) \\ (1.458, 1.491), (1.720, 1.783), (2.098, 2.217) \\ (1.526, 1.637), (1.801, 1.957), (2.196, 2.433) \\ (1.627, 1.745), (1.920, 2.087), (2.342, 2.594) \end{bmatrix}$$

Then, the matrix $IFRN(\Re_j)$ is calculated as follows:

$$IFRN(\Re_6) \begin{bmatrix} \Re_6^{L1} = \left(\frac{\Re_7^{L1}}{\Im_6^{U3}}\right) = \left(\frac{1}{1.852}\right), & \Re_6^{U1} = \left(\frac{\Re_7^{U1}}{\Im_6^{L3}}\right) = \left(\frac{1}{1.549}\right) = (0.540, 0.646) \\ \Re_6^{L2} = \left(\frac{\Re_7^{L2}}{\Im_6^{U2}}\right) = \left(\frac{1}{1.489}\right), & \Re_6^{U2} = \left(\frac{\Re_7^{U2}}{\Im_6^{L2}}\right) = \left(\frac{1}{1.270}\right) = (0.671, 0.787) \\ \Re_6^{L3} = \left(\frac{\Re_7^{L3}}{\Im_6^{U1}}\right) = \left(\frac{1}{1.245}\right), & \Re_6^{U3} = \left(\frac{\Re_7^{U3}}{\Im_6^{L1}}\right) = \left(\frac{1}{1.076}\right) = (0.803, 0.929) \end{bmatrix}$$

The total matrix is as follows:

$$IFRN(\Re_j) = \begin{bmatrix} (1.000, 1.000), (1.000, 1.000), (1.000, 1.000) \\ (0.540, 0.646), (0.671, 0.787), (0.803, 0.929) \\ (0.277, 0.367), (0.429, 0.547), (0.613, 0.761) \\ (0.127, 0.186), (0.244, 0.337), (0.416, 0.554) \\ (0.057, 0.089), (0.137, 0.196), (0.279, 0.380) \\ (0.023, 0.040), (0.070, 0.109), (0.171, 0.249) \\ (0.009, 0.017), (0.033, 0.057), (0.098, 0.153) \end{bmatrix}$$

The sum of the matrix is computed and
$IFRN(\aleph_j) = [(2.034, 2.345), (2.584, 3.033), (3.380, 4.027)]$ is obtained.
Finally,

$$IFRN(W_7) = \left[\left(\frac{1}{4.027}, \frac{1}{3.380}\right), \left(\frac{1}{3.033}, \frac{1}{2.584}\right), \left(\frac{1}{2.345}, \frac{1}{2.034}\right)\right] = [(0.248, 0.296), (0.330, 0.387), (0.426, 0.492)]$$

The final criteria values are shown in Table 4.

**Table 4.** Results of the IFRN SWARA method.

| | wj | | wj |
|---|---|---|---|
| C7 | [(0.248,0.296),(0.330,0.387),(0.426,0.492)] | C1 | [(0.006,0.012),(0.023,0.042),(0.073,0.122)] |
| C6 | [(0.134,0.191),(0.221,0.305),(0.342,0.457)] | C2 | [(0.002,0.005),(0.011,0.022),(0.042,0.075)] |
| C5 | [(0.069,0.109),(0.141,0.212),(0.261,0.374)] | C3 | [(0.014,0.026),(0.045,0.076),(0.119,0.187)] |
| C4 | [(0.031,0.055),(0.080,0.131),(0.178,0.272)] | C4 | [(0.031,0.055),(0.080,0.131),(0.178,0.272)] |
| C3 | [(0.014,0.026),(0.045,0.076),(0.119,0.187)] | C5 | [(0.069,0.109),(0.141,0.212),(0.261,0.374)] |
| C1 | [(0.006,0.012),(0.023,0.042),(0.073,0.122)] | C6 | [(0.134,0.191),(0.221,0.305),(0.342,0.457)] |
| C2 | [(0.002,0.005),(0.011,0.022),(0.042,0.075)] | C7 | [(0.248,0.296),(0.330,0.387),(0.426,0.492)] |

Based on the analysis by experts in this field, using the presented seven criteria (*C1–C7*), the least significant criterion refers to the ascent/descent (*C2*) on the measuring segments that are not shorter than 1000 m before an imagined cross-section and to the volume of traffic (*C1*), which is expressed as the AADT (*veh/day*). The other five criteria (*C3–C7*) represent the arithmetic means (AM) of headways based on the measured values of a total of 12,801 measurements according to vehicle classes from PC-PC to PC-AT. The importance of these five criteria is *C7 > C6 > C5 > C4 > C3*.

*4.4. Determining Overall Efficiency Using the IFRN WASPAS Method*

In this section of the paper, the final efficiency of the observed road sections is determined based on headways. Table 5 shows a group assessment for the first road section.

**Table 5.** The values of the fuzzy numbers for the first road section.

| | DM1 | DM2 | DM3 | DM1 | DM2 | DM3 | DM1 | DM2 | DM3 |
|---|---|---|---|---|---|---|---|---|---|
| | | l | | | m | | | u | |
| C1 | 5 | 5 | 5 | 7 | 7 | 7 | 9 | 9 | 9 |
| C2 | 7 | 7 | 5 | 7 | 9 | 7 | 9 | 9 | 9 |
| C3 | 5 | 7 | 5 | 7 | 9 | 7 | 9 | 9 | 9 |
| C4 | 7 | 7 | 7 | 9 | 9 | 9 | 9 | 9 | 9 |
| C5 | 7 | 9 | 7 | 9 | 9 | 9 | 9 | 9 | 9 |
| C6 | 7 | 7 | 7 | 7 | 7 | 7 | 9 | 7 | 7 |
| C7 | 7 | 7 | 7 | 7 | 9 | 7 | 9 | 9 | 9 |

Then, the rough set values are calculated. The following calculation is based on criterion *C2*:

For *l:*

$$\underline{Lim}(5) = 5, \ \overline{Lim}(5) = \frac{1}{3}(7+7+5) = 6.33; \ \underline{Lim}(7) = \frac{1}{3}(7+7+5) = 6.33, \ \overline{Lim}(7) = 7$$

For *m:*

$$\underline{Lim}(7) = 7, \ \overline{Lim}(7) = \frac{1}{3}(7+9+7) = 7.67; \ \underline{Lim}(9) = \frac{1}{3}(7+9+7) = 7.67, \ \overline{Lim}(9) = 9$$

For *u:*

$$\underline{Lim}(9) = 9, \ \overline{Lim}(9) = 9$$

The first road section is assigned the following interval fuzzy rough numbers for criterion *C2*:

*IFRN (DM$_1$)* = [(6.33, 7.00), (7.00, 7.67), (9.00, 9.00)]

*IFRN (DM$_2$)* = [(6.33, 7.00), (7.67, 9.00), (9.00, 9.00)]

*IFRN (DM$_3$)* = [(5.00, 6,33), (7.00, 7,67), (9,00, 9,00)]

The final value of the interval fuzzy rough numbers is obtained by computing the average values for all DMs. By this approach, an interval fuzzy rough decision matrix is created (Table 6).

**Table 6.** Interval fuzzy rough decision matrix.

| | C1 | C2 | ... | C7 |
|---|---|---|---|---|
| DMU1 | [(5.00,5.00),(7.00,7.00),(9.00,9.00)] | [(5.89,6.78),(7.22,8.11),(9.00,9.00)] | ... | [(7.00,7.00),(7.22,8.11),(9.00,9.00)] |
| DMU3 | [(5.00,5.00),(7.00,7.00),(9.00,9.00)] | [(5.00,5.00),(7.00,7.00),(7.22,8.11)] | ... | [(7.00,7.00),(9.00,9.00),(9.00,9.00)] |
| DMU4 | [(5.00,5.00),(7.00,7.00),(9.00,9.00)] | [(5.00,5.00),(7.00,7.00),(7.00,7.00)] | ... | [(7.00,7.00),(7.00,7.00),(7.22,8.11)] |
| DMU6 | [(7.22,8.11),(9.00,9.00),(9.00,9.00)] | [(3.00,3.00),(3.89,4.78),(7.00,7.00)] | ... | [(5.00,5.00),(7.00,7.00),(7.00,7.00)] |
| DMU9 | [(3.00,3.00),(3.89,4.78),(5.89,6.78)] | [(5.22,6.11),(7.00,7.00),(9.00,9.00)] | ... | [(3.00,3.00),(5.00,5.00),(5.00,5.00)] |
| DMU10 | [(3.00,3.00),(5.00,5.00),(7.00,7.00)] | [(7.00,7.00),(7.00,7.00),(9.00,9.00)] | ... | [(3.00,3.00),(3.00,3.00),(5.00,5.00)] |
| DMU11 | [(5.00,5.00),(7.00,7.00),(9.00,9.00)] | [(5.89,6.78),(7.22,8.11),(9.00,9.00)] | ... | [(7.00,7.00),(7.22,8.11),(9.00,9.00)] |
| Max | [(7.22,8.11),(9.00,9.00),(9.00,9.00)] | [(7.00,7.00),(7.22,8.11),(9.00,9.00)] | ... | [(7.00,7.00),(9.00,9.00),(9.00,9.00)] |

The normalization of the initial IFRN matrix is performed by applying Equation (21) since all criteria have been modeled as a benefit, and it is shown in Table 7. For DMU1, according to the first criterion, it is as follows:

$$d_{DMU1} = \left[ \left( \frac{5.00}{9.00}, \frac{5.00}{9.00} \right), \left( \frac{7.00}{9.00}, \frac{7.00}{9.00} \right), \left( \frac{9.00}{8.10}, \frac{9.00}{7.20} \right) \right] = [(0.56, 0.56), (0.78, 0.78), (1.11, 1.25)]$$

**Table 7.** Normalized interval fuzzy rough decision matrix.

| | C1 | C2 | ... | C7 |
|---|---|---|---|---|
| DMU1 | [(0.56,0.56),(0.78,0.78),(1.11,1.25)] | [(0.65,0.75),(0.89,1.12),(1.29,1.29)] | ... | [(0.78,0.78),(0.80,0.90),(1.29,1.29)] |
| DMU3 | [(0.56,0.56),(0.78,0.78),(1.11,1.25)] | [(0.56,0.56),(0.86,0.97),(1.03,1.16)] | ... | [(0.78,0.78),(1.00,1.00),(1.29,1.29)] |
| DMU4 | [(0.56,0.56),(0.78,0.78),(1.11,1.25)] | [(0.56,0.56),(0.86,0.97),(1.00,1.00)] | ... | [(0.78,0.78),(0.78,0.78),(1.03,1.16)] |
| DMU6 | [(0.80,0.90),(1.00,1.00),(1.11,1.25)] | [(0.33,0.33),(0.48,0.66),(1.00,1.00)] | ... | [(0.56,0.56),(0.78,0.78),(1.00,1.00)] |
| DMU9 | [(0.33,0.33),(0.43,0.53),(0.73,0.94)] | [(0.58,0.68),(0.86,0.97),(1.29,1.29)] | ... | [(0.33,0.33),(0.56,0.56),(0.71,0.71)] |
| DMU10 | [(0.33,0.33),(0.56,0.56),(0.86,0.97)] | [(0.78,0.78),(0.86,0.97),(1.29,1.29)] | ... | [(0.33,0.33),(0.33,0.33),(0.71,0.71)] |
| DMU11 | [(0.56,0.56),(0.78,0.78),(1.11,1.25)] | [(0.65,0.75),(0.89,1.12),(1.29,1.29)] | ... | [(0.78,0.78),(0.80,0.90),(1.29,1.29)] |

After that, the weighting of the normalized decision matrix is completed, multiplying the normalized data by corresponding weights. For the previous examples, the computation procedure is as follows:

$$v_{DMU1} = [(0.56 \times 0.006, 0.56 \times 0.012), (0.78 \times 0.023, 0.78 \times 0.042), (1.11 \times 0.073, 1.25 \times 0.122)]$$
$$= [(0.003, 0.07), (0.018, 0.033), (0.081, 0.153)]$$

Then, the function $IFRN(S_{ij})$ is computed as follows:

$$IFRN(S_{DMU1}) = \begin{bmatrix} (0.00 + 0.00 + 0.01 + 0.02 + 0.06 + 0.10 + 0.19), (0.01 + 0.00 + 0.02 + 0.04 + 0.10 + 0.15 + 0.23), \\ (0.02 + 0.01 + 0.04 + 0.08 + 0.14 + 0.18 + 0.26), (0.03 + 0.02 + 0.08 + 0.13 + 0.21 + 0.27 + 0.35), \\ (0.08 + 0.05 + 0.15 + 0.23 + 0.29 + 0.35 + 0.55), (0.15 + 0.10 + 0.24 + 0.35 + 0.47 + 0.53 + 0.63) \end{bmatrix} =$$
$$[(0.39, 0.55), (0.73, 1.10), (1.71, 2.47)]$$

and after that, the function $IFRN(Q_{ij})$ is as follows:

$$IFRN(Q_{DMU1}) = \begin{bmatrix} \left( (0.56)^{0.122} \times (0.65)^{0.075} \times (0.58)^{0.187} \times (0.78)^{0.272} \times (0.80)^{0.374} \times (0.78)^{0.457} \times (0.78)^{0.492} \right) \\ \left( (0.56)^{0.073} \times (0.75)^{0.042} \times (0.68)^{0.119} \times (0.78)^{0.178} \times (0.90)^{0.261} \times (0.78)^{0.342} \times (0.78)^{0.426} \right) \\ \left( (0.78)^{0.042} \times (0.89)^{0.022} \times (0.82)^{0.076} \times (1.00)^{0.131} \times (1.00)^{0.212} \times (0.80)^{0.305} \times (0.80)^{0.387} \right) \\ \left( (0.78)^{0.023} \times (1.12)^{0.011} \times (1.03)^{0.045} \times (1.00)^{0.080} \times (1.00)^{0.141} \times (0.89)^{0.221} \times (0.90)^{0.330} \right) \\ \left( (1.11)^{0.012} \times (1.29)^{0.005} \times (1.29)^{0.026} \times (1.29)^{0.055} \times (1.11)^{0.109} \times (1.03)^{0.191} \times (1.29)^{0.296} \right) \\ \left( (1.25)^{0.006} \times (1.29)^{0.002} \times (1.29)^{0.014} \times (1.29)^{0.031} \times (1.25)^{0.069} \times (1.16)^{0.134} \times (1.29)^{0.248} \right) \end{bmatrix} = [(0.74, 0.86), (0.96, 0.99), (1.01, 1.06)]$$

Finally, the alternatives are ranked in descending order based on the obtained final values:

$$IFRN(T_{DMU1}) = \begin{bmatrix} (0.50 \times 0.39 + 0.50 \times 0.74), (0.50 \times 0.55 + 0.50 \times 0.86), (0.50 \times 0.73 + 0.50 \times 0.96) \\ (0.50 \times 1.10 + 0.50 \times 0.99), (0.50 \times 1.71 + 0.50 \times 1.01), (0.50 \times 2.47 + 0.50 \times 1.06) \end{bmatrix} = [(0.57, 0.71), (0.85, 1.05), (1.36, 1.76)]$$

The final results after applying the integrated DEA-IFRN SWARA-IFRN WASPAS model are given in Table 8.

**Table 8.** Results of applying the DEA-IFRN SWARA-IFRN WASPAS model.

| | $IFRN\left(S_{ij}\right)$ | $IFRN\left(Q_{ij}\right)$ | $IFRN\left(T_{ij}\right)$ | Rank |
|---|---|---|---|---|
| DMU1 | [(0.39,0.548),(0.728,1.097),(1.707,2.468)] | [(0.743,0.864),(0.962,0.994),(1.013,1.057)] | 1.05 | 1 |
| DMU3 | [(0.384,0.529),(0.768,1.14),(1.733,2.438)] | [(0.706,0.822),(0.951,0.985),(1.01,1.069)] | 1.04 | 2 |
| DMU4 | [(0.297,0.401),(0.52,0.794),(1.295,1.893)] | [(0.538,0.717),(0.874,0.949),(0.992,0.976)] | 0.85 | 4 |
| DMU6 | [(0.292,0.406),(0.682,1.004),(1.563,2.233)] | [(0.664,0.791),(0.935,0.981),(1.009,1.006)] | 0.96 | 3 |
| DMU9 | [(0.163,0.228),(0.462,0.677),(1.093,1.58)] | [(0.395,0.607),(0.843,0.928),(0.984,0.918)] | 0.74 | 5 |
| DMU10 | [(0.122,0.174),(0.279,0.443),(1.014,1.501)] | [(0.333,0.542),(0.789,0.894),(0.979,0.917)] | 0.67 | 6 |
| DMU11 | [(0.07,0.124),(0.324,0.517),(0.902,1.309)] | [(0.314,0.529),(0.819,0.912),(0.976,0.86)] | 0.64 | 7 |

By analyzing the ranking of efficiency, and applying the DEA-IFRN SWARA-IFRN WASPAS model, the ranking of alternatives for the observed sections of two-lane roads was performed. According to the results obtained, the best-ranked is a measuring segment of the Ivanjska–Šargovac section, with a road gradient = −5.5%, which has low deviating values of headways according to the measurement classes from PC-PC to AT-PC, which shows balanced and continuous traffic flows. The Vrhovi–Šešlije section, with a road gradient = 5%, stands out as the worst-ranked of the given sections, where the headway values from PC-PC to AT-PC differ by up to 20 times. It is obvious that the measure of efficiency, which refers to continuous traffic flows on this section, was significantly lost by applying the given model.

## 5. Verification Tests

### 5.1. Sensitivity Analysis (SA)

In order to be able to determine the stability of the obtained results and the influence of criterion values on the final ranks of the alternatives, a sensitivity analysis is often performed, which has been confirmed by a number of studies [35–38]. In this section, 70 new cases have been formed with new values of seven criteria, whereby their values have been reduced to within a range of 5–95%. The values of all criteria are shown in Figure 1.

The next step entails the creation of 70 new IFRN SWARA-IFRN WASPAS models by implementing new simulated criteria values in each scenario. The results of changing the criteria values are given in Figure 2.

The ranks obtained through the 70 scenarios show the stability of the model and confirm the initial results, regardless of the change to the best alternative. In general, as previously noted, DMU1 and DMU3 represent road sections that are almost identical in terms of final efficiency. Therefore, it is not surprising that they exchanged their positions in certain scenarios, primarily due to the drastic drop in the value of the first, fourth, and fifth criteria. There were a total of 17 such cases, which is 24.29%.

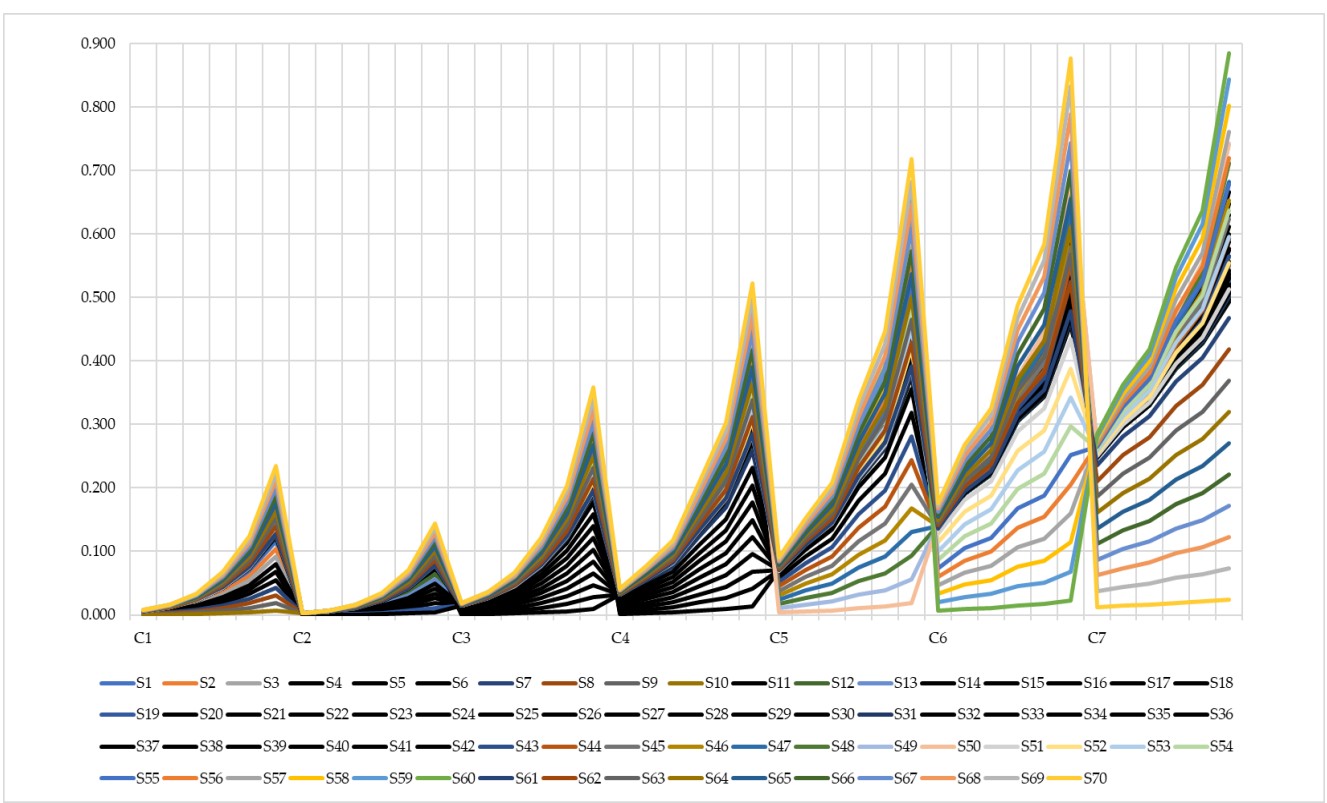

**Figure 1.** New criteria weights in 70 cases.

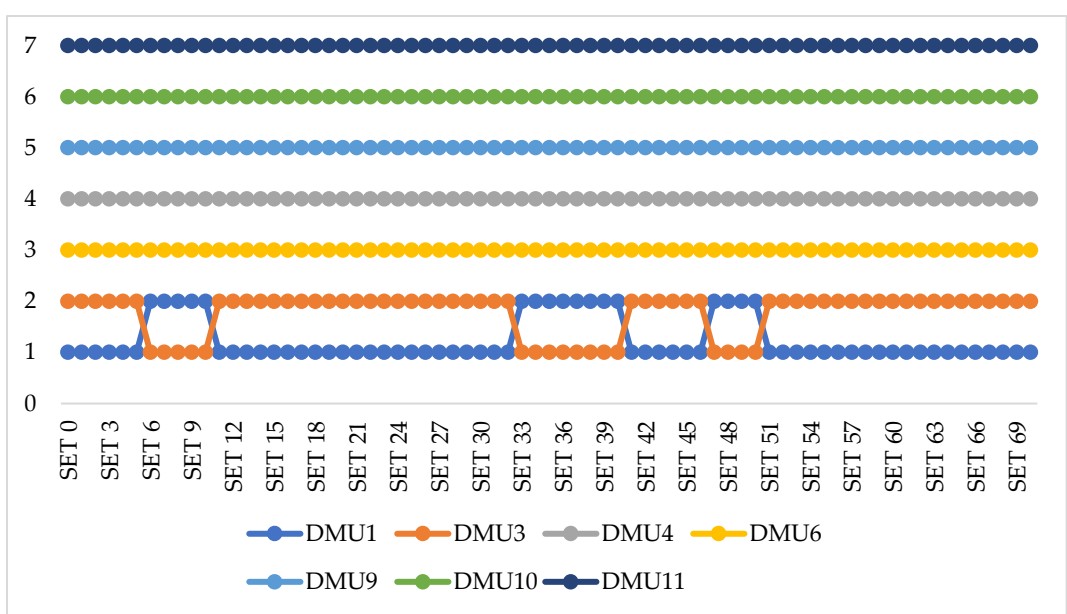

**Figure 2.** Results of the SA.

*5.2. Comparative Analysis (CA)*

This section refers to the application of four other MCDM methods—MARCOS: measurement of alternatives and ranking according to the compromise solution [39]; MABAC: multi-attributive border-approximation area comparison [35]; GRADIS: compromise ranking of alternatives from distance to ideal solution [40]; and ARAS: additive ratio assessment [41]—in the IFRN environment in order to confirm the new results (Figure 3) and compliance with the initial ranks of the IFRN SWARA-IFRN WASPAS model.

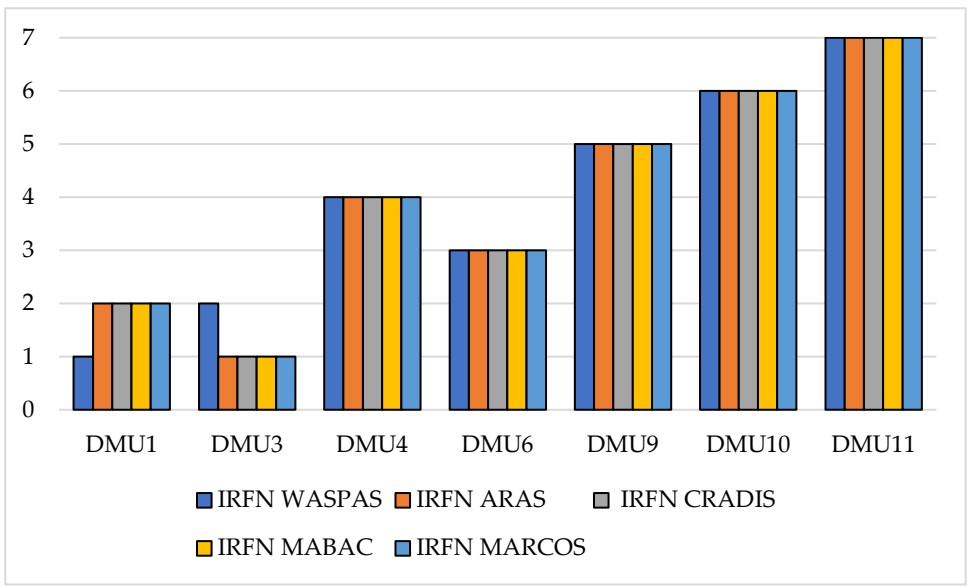

**Figure 3.** Ranks in the CA.

The CA results confirm what has been presented in the SA, i.e., DMU3 has the highest efficiency and is ranked first. This is also the case with the IFRN WASPAS method at a higher value of the $\lambda$ parameter, which is verified in the next section.

### 5.3. Changing the $\lambda$ Parameter

An integral part of the IFRN WASPAS method is the coefficient $\lambda$, with a range of 0–1, where its mean value, i.e., 0.50, is most often taken. In this section of the paper, the influence of this coefficient on the ranks of road sections has been determined, which is shown in Figure 4.

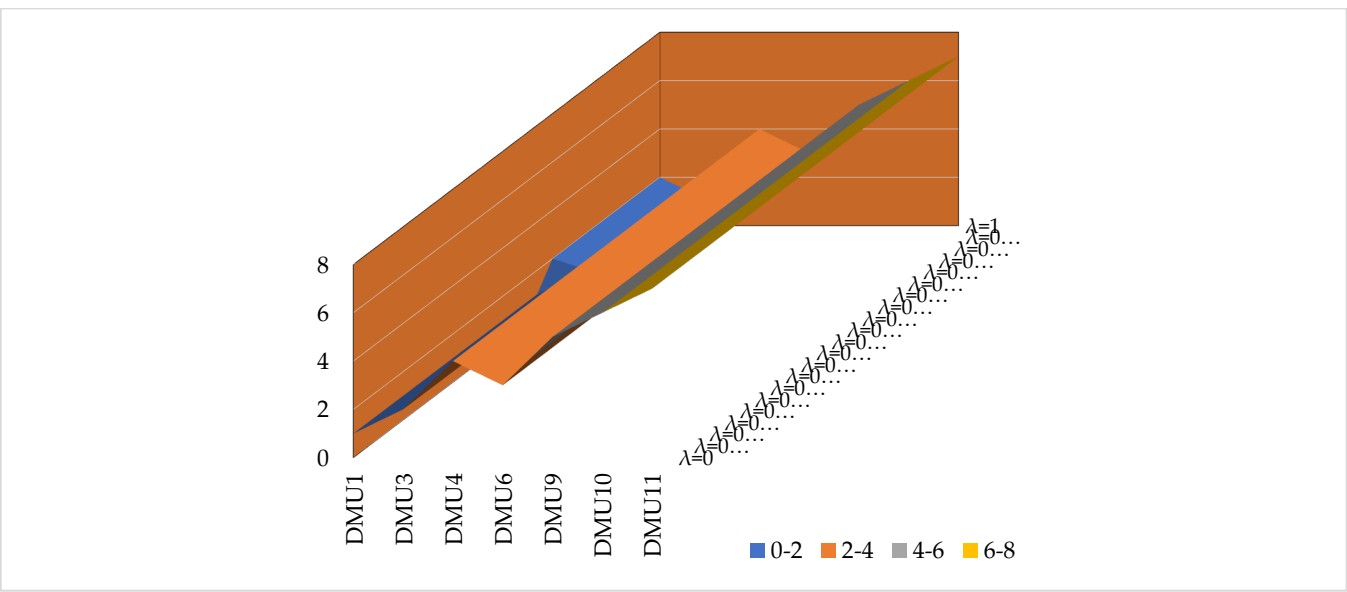

**Figure 4.** Ranking of road sections in accordance to changing the $\lambda$ parameter.

The parameter $\lambda$ was modeled in its established interval of 0–1 with a sequence of 0.05, which means that a total of 21 scenarios were formed, including the value of 0.50. The results show that at a value of this parameter of up to 0.60, DMUs keep their original positions, while at a value of 0.65–1.00, DMU3 becomes the most efficient section, which is also the case in the CA.

### 5.4. Reverse Rank (RR) Analysis

In order to ensure the credibility of the proposed IFRN SWARA-IFRN WASPAS model, a reverse-rank analysis [42] with different variations was performed. First, different scenarios were formed, implying that the worst road section was eliminated. After that, the worst alternative was added to the existing structure of the initial interval fuzzy-rough-number matrix. The next scenario implied that the worst alternative was replaced with the second worst, and in the last scenario, the two most significant criteria (*C7* and *C6*) were removed from the model. The results from the aspect of reverse rank analysis with the values of road sections are shown in Figure 5, i.e., from the aspect of ranks in Figure 6.

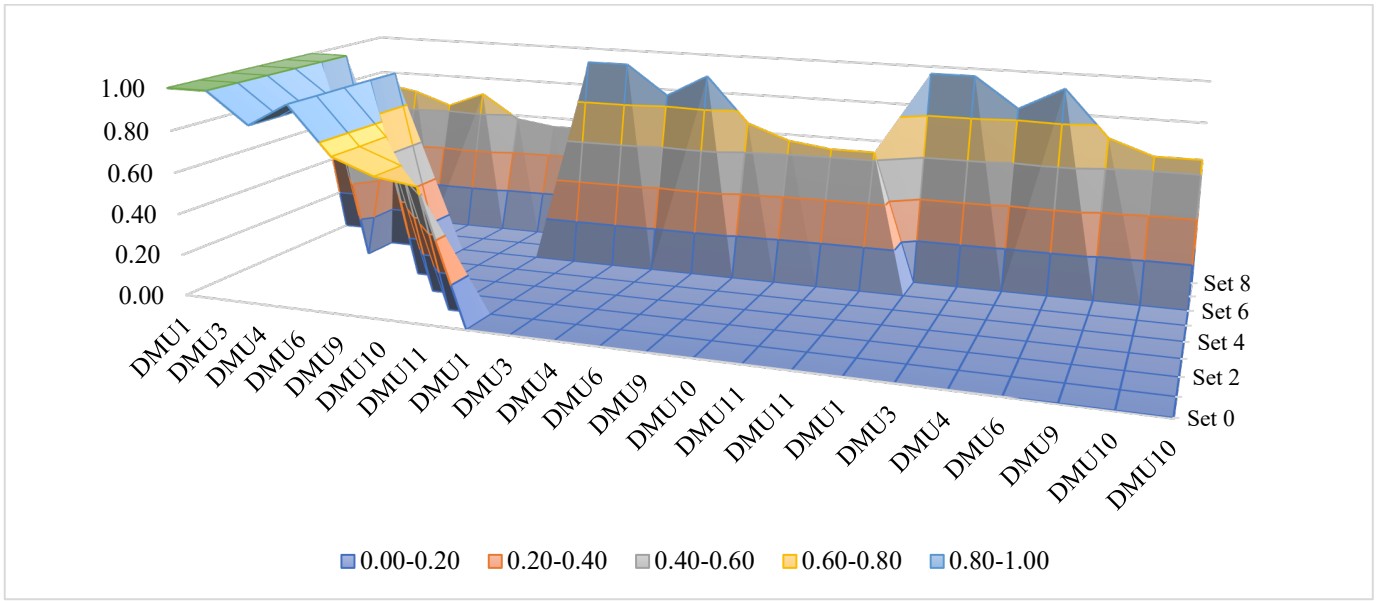

**Figure 5.** Values of road sections after the RR analysis.

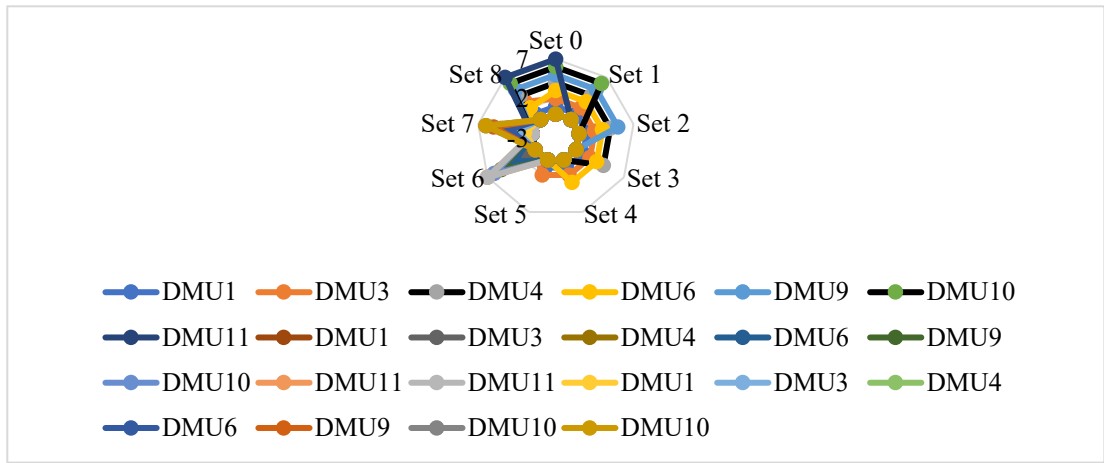

**Figure 6.** Ranks of road sections after the RR analysis.

In general, the results concerning the change in the final values of the alternatives in the RR analysis do not show large deviations compared to the original values.

When it comes to ranks, DMU1 remains the most efficient road section in the overall RR analysis. In addition, the other DMUs keep their original positions, except in the last scenario, where the two most influential criteria are deleted. Then, DMU3 and DMU6 exchange their positions.

## 6. Conclusions

Based on the development of the model and its application in a specific case study of 14 initial measuring segments of the given sections, ranking from the aspect of the efficiency of traffic flows on the sections was conducted. By generalizing and modeling the applied criteria of AADT, gradients, and AM headways, it was shown that the rank of efficiency is achieved with a lower deviation of headways according to the specified vehicle class. Also, the measure of efficiency implies an optimal flow, balanced and continuous, which is especially noticeable for the measuring segment of the Ivanjska–Šargovac section on a descent of $-5.5\%$, where there is a low headway deviation. Too-high headway deviations imply an unbalanced flow, and these are often noticeable on an ascent, which caused some of those sections to be ranked significantly worse in terms of efficiency measures. Using a specific real case study that contains 14 road sections gives practical relevance to the research. By analyzing real data and using the proposed model to evaluate efficiency based on headway analysis, the research provides actionable insights for improving traffic flow and infrastructure management.

In this paper, a novel integrated DEA-IFRN SWARA-IFRN WASPAS model has been developed to determine the efficiency of 14 road infrastructure sections. The contribution of the research can be viewed in two ways, from a scientific–methodological and professional aspect. From the scientific aspect, it is certainly a dominant contribution, which is reflected in the development of the IFRN WASPAS method, while from the professional aspect, it represents support for the infrastructure manager and traffic experts in order to define certain measures. Also, the proposed innovative model has the potential to advance the field of transportation engineering by providing a more comprehensive analysis of infrastructure efficiency.

On the other hand, the findings of the study may have limited generalizability due to the focus on a specific set of road sections and the relatively small sample size. This restricts the broader applicability of the developed model to different geographical contexts or transportation systems, potentially limiting its usefulness to practitioners in diverse settings. Limitations related to this research can be manifested through the relatively short measuring segments (with a length of 1000 m) and the small number of DMs who participated in the group decision-making. Also, one of the limitations may be the lack of new data related to the AADT or the fact that integration of multiple decision-making methods and the use of interval fuzzy rough numbers may introduce methodological complexity, making it challenging for readers to grasp the intricacies of the approach. This complexity could hinder understanding and replication by other researchers or practitioners, potentially limiting the adoption of the developed model. These limitations can be mitigated if the reproduction of the model is made soon with more parameters and if the model is applied under the advice of experts in the field of methodology.

Future research refers to the collection of data and the determination of the efficiency of new sections of road infrastructure, as well as the definition of additional parameters and the inclusion of DMs for different structures. The developed model can be applied in any other case study that contains multiple variants and criteria. Also, from a methodological aspect, future research can be related to extension methods in other forms like quasirung fuzzy sets [43], polytopic fuzzy sets [44], integration with machine learning [45,46], multi-objective optimization [47], etc.

**Author Contributions:** Conceptualization, M.S. and D.A.; methodology, Ž.S. and G.S.; validation, D.K.D. and M.S.; formal analysis, N.N. and M.S.; investigation, D.A. and Ž.S.; writing—original draft preparation, Ž.S., D.A. and G.S.; writing—review and editing, D.K.D., N.N. and M.S.; visualization, N.N. All authors have read and agreed to the published version of the manuscript.

**Funding:** This research has been supported by the Ministry of Science, Technological Development and Innovation (Contract No. 451-03-65/2024-03/200156) and the Faculty of Technical Sciences, University of Novi Sad through project "Scientific and Artistic Research Work of Researchers in Teaching and Associate Positions at the Faculty of Technical Sciences, University of Novi Sad" (No. 01-3394/1).

**Data Availability Statement:** The data presented in this study are available on request from the corresponding author.

**Acknowledgments:** The paper is part of the research conducted within Project No. 19.032/961-39/23 "Investigation of time headway in models of efficiency and safety analysis of two-lane roads" supported by the Ministry of Scientific and Technological Development and Higher Education of the Republic of Srpska.

**Conflicts of Interest:** The authors declare no conflicts of interest.

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
