# Peer review of "A Novel Data-Envelopment Analysis Interval-Valued Fuzzy-Rough-Number Multi-Criteria Decision-Making (DEA-IFRN MCDM) Model for Determining the Efficiency of Road Sections Based on Headway Analysis"

_mathematics, doi:10.3390/math12070976_

Round 1

Reviewer 1 Report

Comments and Suggestions for Authors

 This paper develops a novel interval fuzzy rough number decision-making model consisting of DEA, SWARA, and WASPAS methods.  The paper under consideration explores a matter of great interest and importance in the field of decision making. Overall, the paper is well-written and organized, and the methodology is adequately described. However, I would suggest a series of changes that, in my opinion, would improve the paper further.

1.      I suggest revising the paper title to make it more concise and suitable. The repeating term IFRN makes it wordy.

2.      It is recommended that the abstract be carefully rewritten to effectively demonstrate the necessity, novelty, and contribution of the research, as well as highlight its major findings.

3.      There are too many abbreviations without proper definitions. It creates unnecessary confusion. Acronyms should be properly defined. The main contribution of the manuscript should be listed in bullets under the Introduction section

4.      The introduction section will have to improve given the research gaps.  Why interval fuzzy rough numbers are chosen as the research topic? What is the practical relevance of this study? It should be described.

5.      Why only the SWARA method is used for weight determination? Nowadays authors utilize a combination of subjective and objective methods. What are the advantages? Please refer to and cite to improve this part ‘Determination of best renewable energy sources in India using SWARA-ARAS in confidence level based interval-valued Fermatean fuzzy environment’.

6.      Result analysis and comparative sections are well written.

7.      It is imperative to explain the theoretical and practical implications of this research.

8.      The motivation for the prospects section in the manuscript is not clearly conveyed. Your method could be applied to another extension of fuzzy sets such as ‘quasirung fuzzy sets’. Include these in the future scope of the conclusion section by properly citing relevant references. 

Author Response

Reviewer 1:

Thank you very much for the useful suggestions. We accepted all of the suggestions and we are sure that this will improve the quality and contribute to a better understanding of the paper.

This paper develops a novel interval fuzzy rough number decision-making model consisting of DEA, SWARA, and WASPAS methods.  The paper under consideration explores a matter of great interest and importance in the field of decision making. Overall, the paper is well-written and organized, and the methodology is adequately described. However, I would suggest a series of changes that, in my opinion, would improve the paper further.

 ------------------------------------------------------------------------------------------------------------

Comment 1: I suggest revising the paper title to make it more concise and suitable. The repeating term IFRN makes it wordy.

Reply: Thank you for your comment, you have right. We have changed the title of the paper. Now is: A Novel DEA-IFRN MCDM model for efficiency of road sections based on headway analysis.

Comment 2: It is recommended that the abstract be carefully rewritten to effectively demonstrate the necessity, novelty, and contribution of the research, as well as highlight its major findings.

Reply:

Novelty and contributions: We have developed a novel interval fuzzy rough number decision-making model consisting of DEA (Data envelopment analysis), IFRN SWARA (Interval-valued fuzzy-rough number Stepwise Weight Assessment Ratio Analysis), and IFRN WASPAS (Interval-valued fuzzy-rough number Weighted Aggregates Sum Product Assessment) methods. The main contribution of this study is a new extension of WASPAS method with interval fuzzy rough numbers.

Results: Results show that two sections are very similar and have almost equal efficiency, while other results are very stable. According to the results obtained, the best-ranked is a measuring segment of the Ivanjska-Šargovac section with a road gradient=-5.5%, which has low deviating values of headways according to the measurement classes from PC-PC to AT-PC, which shows balanced and continuous traffic flows.

Comment 3: There are too many abbreviations without proper definitions. It creates unnecessary confusion. Acronyms should be properly defined. The main contribution of the manuscript should be listed in bullets under the Introduction section.

Reply: Each abbreviation has been defined at the first place of appearance. Contributions have been listed in the introduction section:

  • In accordance with the previously defined importance of headways for the entire area of road transport, a total of 14 sections of road infrastructure were consid-ered and a new model to determine their efficiency was created. In addition to five headway classifications, AADT (Annual average daily traffic) and road gra-dient were taken as influential parameters.
  • A new multiphase efficiency model, which includes the DEA model, SWARA and WASPAS methods in the form of interval fuzzy rough numbers, has been created. IFRNs have been used due to their ability to treat uncertainty in the decision-making process adequately. The greatest contribution of the paper can be seen from the aspect of a new algo-rithm of the IFRN WASPAS method, which, according to the authors' knowledge, is presented for the first time in the literature. So far, certain comparative anal-yses have been presented, but without the algorithm of this method.
  • Another aspect of contribution is reflected through the sustainable management of road infrastructure based on the results obtained and future recommendations. On the practical aspect, the study provides valuable insights for infrastructure managers and traffic experts, helping them make informed decisions to optimize road section efficiency.

Comment 4: The introduction section will have to improve given the research gaps.  Why interval fuzzy rough numbers are chosen as the research topic? What is the practical relevance of this study? It should be described.

Reply: We have added the following in the introduction section.

Another aspect of contribution is reflected through the sustainable management of road infrastructure based on the results obtained and future recommendations. On the practical aspect, the study provides valuable insights for infrastructure managers and traffic experts, helping them make informed decisions to optimize road section efficiency.

Research gaps can be described in the following. The paper thoroughly analyzes road section efficiency by considering multiple input-output parameters, determining criteria weights by applying the IFRN SWARA method, sorting road sections with the IFRN WASPAS method, and performing verification tests. This comprehensive model enhances the credibility and reliability of the research findings.

IFRNs have been used due to their ability to treat uncertainty in the decision-making process adequately.

Comment 5: Why only the SWARA method is used for weight determination? Nowadays authors utilize a combination of subjective and objective methods. What are the advantages? Please refer to and cite to improve this part ‘Determination of best renewable energy sources in India using SWARA-ARAS in confidence level based interval-valued Fermatean fuzzy environment’.

Reply: We have used this subjective method for calculating criteria weights because this case study needs adequate expertise of DMs and their preferences based on skills and knowledge. This method is extended with numerous theories [27-30] and finally with IFRN [31].

Comment 6: Result analysis and comparative sections are well written.

Reply: Thank you for comment.

Comment 7: It is imperative to explain the theoretical and practical implications of this research.

Reply: Using a specific real case study that contains 14 road sections gives practical relevance to the research. Analyzing real data and using the proposed model to evaluate efficiency based on headway analysis, the research provides actionable insights for improving traffic flow and infrastructure management. On the other hand, the findings of the study may have limited generalizability due to the focus on a specific set of road sections and the relatively small sample size. This restricts the broader applicability of the developed model to different geographical contexts or transportation systems, potentially limiting its usefulness to practitioners in diverse settings.

Comment 8: The motivation for the prospects section in the manuscript is not clearly conveyed. Your method could be applied to another extension of fuzzy sets such as ‘quasirung fuzzy sets’. Include these in the future scope of the conclusion section by properly citing relevant references. 

Reply: Yes, we have added it in the conclusion section as follows: Also, from a methodological aspect future research can be related to extension methods in other forms like quasirung fuzzy sets [43], polytopic fuzzy sets [44], integration with machine learning [45,46], multi-objective optimization [47] etc.

Reviewer 2 Report

Comments and Suggestions for Authors

The research addresses a crucial aspect of transportation engineering, namely the assessment of road section efficiency. Given the significant impact of road infrastructure on transportation systems and the economy, the study's focus on optimizing efficiency is highly relevant and timely.The integration of DEA, IFRN SWARA, and IFRN WASPAS methods into a single decision-making model represents a novel approach to evaluating the efficiency of road sections. This innovative methodology has the potential to advance the field of transportation engineering by providing a more comprehensive and nuanced analysis of infrastructure efficiency.

The article provides a thorough analysis of road section efficiency by considering multiple input-output parameters, calculating final weights using the IFRN SWARA method, ranking road sections with the IFRN WASPAS method, and conducting verification tests. This comprehensive approach enhances the credibility and reliability of the research findings.

The study offers both scientific and practical contributions to the field. On the scientific side, the development of the IFRN WASPAS method represents a significant advancement in decision-making models. On the practical side, the research provides valuable insights for infrastructure managers and traffic experts, helping them make informed decisions to optimize road section efficiency.

The use of a specific case study involving 14 road sections adds practical relevance to the research. By analyzing real-world data and applying the developed model to assess efficiency measures such as headway deviation, the study provides actionable insights for improving traffic flow and infrastructure management.

On the other hand, the findings of the study may have limited generalizability due to the focus on a specific set of road sections and the relatively small sample size. This restricts the broader applicability of the developed model to different geographical contexts or transportation systems, potentially limiting its usefulness to practitioners in diverse settings.

The integration of multiple decision-making methods and the use of interval fuzzy rough numbers may introduce methodological complexity, making it challenging for readers to grasp the intricacies of the approach. This complexity could hinder understanding and replication by other researchers or practitioners, potentially limiting the adoption of the developed model.

There is a notable disconnect between the content described in the abstract, which focuses on road section efficiency based on headway analysis, and the conclusion, which discusses the development of the IFRN WASPAS method. This inconsistency undermines the coherence of the paper and may confuse readers about the primary focus and contributions of the research.

 While the article acknowledges limitations related to short measuring segments and a small number of decision-makers, it does not provide sufficient discussion on how these limitations were mitigated or their potential impact on the research findings. Providing more transparency about data limitations and their implications would enhance the credibility and rigor of the study.

The article lacks a discussion on the external validity of the findings and the potential transferability of the developed model to other settings or contexts. Addressing this aspect would strengthen the research by providing insights into the applicability of the model beyond the specific case study considered.

Overall, while the article presents an innovative approach to assessing road section efficiency, there are notable weaknesses related to limited generalizability, methodological complexity, coherence, and data limitations that need to be addressed to improve the quality and impact of the research.

Author Response

Reviewer 2:

Thank you very much for the useful suggestions. We accepted all of the suggestions and we are sure that this will improve the quality and contribute to a better understanding of the paper.

The research addresses a crucial aspect of transportation engineering, namely the assessment of road section efficiency. Given the significant impact of road infrastructure on transportation systems and the economy, the study's focus on optimizing efficiency is highly relevant and timely. The integration of DEA, IFRN SWARA, and IFRN WASPAS methods into a single decision-making model represents a novel approach to evaluating the efficiency of road sections. This innovative methodology has the potential to advance the field of transportation engineering by providing a more comprehensive and nuanced analysis of infrastructure efficiency.

The article provides a thorough analysis of road section efficiency by considering multiple input-output parameters, calculating final weights using the IFRN SWARA method, ranking road sections with the IFRN WASPAS method, and conducting verification tests. This comprehensive approach enhances the credibility and reliability of the research findings.

The study offers both scientific and practical contributions to the field. On the scientific side, the development of the IFRN WASPAS method represents a significant advancement in decision-making models. On the practical side, the research provides valuable insights for infrastructure managers and traffic experts, helping them make informed decisions to optimize road section efficiency.

The use of a specific case study involving 14 road sections adds practical relevance to the research. By analyzing real-world data and applying the developed model to assess efficiency measures such as headway deviation, the study provides actionable insights for improving traffic flow and infrastructure management.

Reply: Thank you for your positive opinion and extensive elaboration of the paper.

------------------------------------------------------------------------------------------------------------

Comment 1: On the other hand, the findings of the study may have limited generalizability due to the focus on a specific set of road sections and the relatively small sample size. This restricts the broader applicability of the developed model to different geographical contexts or transportation systems, potentially limiting its usefulness to practitioners in diverse settings.

Reply: Thank you. We have added your comment in conclusion section.

Comment 2: The integration of multiple decision-making methods and the use of interval fuzzy rough numbers may introduce methodological complexity, making it challenging for readers to grasp the intricacies of the approach. This complexity could hinder understanding and replication by other researchers or practitioners, potentially limiting the adoption of the developed model.

Reply: Yes, you gave a right. We have added your comment in conclusion section.

Comment 3:  There is a notable disconnect between the content described in the abstract, which focuses on road section efficiency based on headway analysis, and the conclusion, which discusses the development of the IFRN WASPAS method. This inconsistency undermines the coherence of the paper and may confuse readers about the primary focus and contributions of the research.

Reply: We have slightly modified the abstract, while the conclusion section has been extended and modified.

Abstract: The capacity of transport infrastructure is one of very important tasks in transport engineering, depending mostly on the geometric characteristics of road and headway analysis. In this paper, we have considered 14 road sections and determined their efficiency based on headway analysis. We have developed a novel interval fuzzy rough number decision-making model consisting of DEA (Data envelopment analysis), IFRN SWARA (Interval-valued fuzzy-rough number Stepwise Weight Assessment Ratio Analysis), and IFRN WASPAS (Interval-valued fuzzy-rough number Weighted Aggregates Sum Product Assessment) methods. The main contribution of this study is a new extension of WASPAS method with interval fuzzy rough numbers. Firstly, the DEA model has been applied to determine the efficiency of 14 road sections according to seven input-output parameters. Seven out of 14 alternatives show full efficiency and have been implemented further in the model. After that, the IFRN SWARA method has been used for the calculation of final weights, while IFRN WASPAS has been applied for ranking seven road sections. Results show that two sections are very similar and have almost equal efficiency, while other results are very stable. According to the results obtained, the best-ranked is a measuring segment of the Ivanjska-Šargovac section with a road gradient=-5.5%, which has low deviating values of headways according to the measurement classes from PC-PC to AT-PC, which shows balanced and continuous traffic flows. Finally, verification tests such as changing criteria weights, comparative analysis, changing parameter λ, and reverse rank analysis have been performed.

Conclusion:

Based on the development of the model and its application in a specific case study on 14 initial measuring segments of the given sections, ranking from the aspect of the efficiency of traffic flows on the sections was conducted. By generalizing and modeling the applied criteria of AADT, gradients and AM headways, it was shown that the rank of efficiency is achieved with a lower deviation of headways according to the specified vehicle classes. Also, the measure of efficiency implies an optimal flow, balanced and continuous, which is especially noticeable for the measuring segment of the Ivanjska-Šargovac section on a descent of - 5.5% where there is a low headway devia-tion. Too high headway deviations imply an unbalanced flow, and are often noticeable on an ascent, which caused some of those sections to be ranked significantly worse in terms of efficiency measures. Using a specific real case study that contains 14 road sec-tions gives practical relevance to the research. Analyzing real data and using the pro-posed model to evaluate efficiency based on headway analysis, the research provides actionable insights for improving traffic flow and infrastructure management.

In this paper, a novel integrated DEA-IFRN SWARA-IFRN WASPAS model has been developed to determine the efficiency of 14 road infrastructure sections. The con-tribution of the research can be viewed in two ways, from a scientific-methodological and professional aspect. From the scientific aspect, it is certainly a dominant contribu-tion, which is reflected in the development of the IFRN WASPAS method, while from the professional aspect, it represents support for the infrastructure manager and traffic experts in order to define certain measures. Also, the proposed innovative model has the potential to advance the field of transportation engineering by providing a more comprehensive analysis of infrastructure efficiency.

On the other hand, the findings of the study may have limited generalizability due to the focus on a specific set of road sections and the relatively small sample size. This restricts the broader applicability of the developed model to different geographical contexts or transportation systems, potentially limiting its usefulness to practitioners in diverse settings. Limitations related to this research can be manifested through rela-tively short measuring segments (with a length of 1000 meters) and a small number of DMs who participated in group decision-making. Also, one of the limitations may be the lack of new data related to AADT or the fact that integration of multiple deci-sion-making methods and the use of interval fuzzy rough numbers may introduce methodological complexity, making it challenging for readers to grasp the intricacies of the approach. This complexity could hinder understanding and replication by other researchers or practitioners, potentially limiting the adoption of the developed model. These limitations can be mitigated if the reproduction of the model will be made soon with more parameters and if the model will be applied under the advice of experts in the field of methodology.

Future research refers to the collection of data and the determination of the effi-ciency of new sections of road infrastructure, as well as the definition of additional parameters and the inclusion of DMs of different structures. The developed model can be applied in any other case study that contains multiple variants and criteria. Also, from a methodological aspect future research can be related to extension methods in other forms like quasirung fuzzy sets [43], polytopic fuzzy sets [44], integration with machine learning [45,46], multi-objective optimization [47] etc.

Comment 4:  While the article acknowledges limitations related to short measuring segments and a small number of decision-makers, it does not provide sufficient discussion on how these limitations were mitigated or their potential impact on the research findings. Providing more transparency about data limitations and their implications would enhance the credibility and rigor of the study.

Reply: Yes, it is true, we have added the following sentence related to your comment. These limitations can be mitigated if the reproduction of the model will be made soon with more parameters and if the model will be applied under the advice of experts in the field of methodology.

Comment 5:   The article lacks a discussion on the external validity of the findings and the potential transferability of the developed model to other settings or contexts. Addressing this aspect would strengthen the research by providing insights into the applicability of the model beyond the specific case study considered.

Reply: Generally, we have performed extensive external analysis represented in the fifth section. Also, one sentence related to this has been added in conlclusion: The developed model can be applied in any other case study that contains multiple variants and criteria.

Comment 6: Overall, while the article presents an innovative approach to assessing road section efficiency, there are notable weaknesses related to limited generalizability, methodological complexity, coherence, and data limitations that need to be addressed to improve the quality and impact of the research.

Reply: Thank you for your comment. We have adopted your suggestions and noted it in conclusion section.

Reviewer 3 Report

Comments and Suggestions for Authors

The authors present research on “A Novel DEA-IFRN SWARA-IFRN WASPAS model for efficiency of road sections based on headway analysis” is interesting work and study begin from the development of a novel interval fuzzy rough number decision-making model consisting of DEA, IFRN SWARA, and IFRN WASPAS methods. The contribution covers the study is a new extension of WASPAS method with interval fuzzy rough numbers. However, there are some questions remaining to be explained and the manuscript should be updated according to the follows.

Following observation for improvement:

1)      The abstract should be structured with background of WASPAS model and verification based on fuzzy model followed by different evaluation methods, key factors that influence accuracy in these different schemes and briefly explain key findings and conclusion for better clarity. Refer Electric vehicle battery capacity degradation and health estimation using machine-learning techniques: a review.

2)      In the introduction section, different input and system operating conditions of fuzzy model schemes and brand concepts on leading vehicle and on its vehicle-dynamic characteristic and verification schemes are not clearly stated. Refer analyzing electric vehicle battery health performance using supervised machine learning.

3)      In the introduction section objective and methodology must be added for better understanding, reproducibility. Look at “a comprehensive review of categorization and perspectives on state-of-charge estimation using deep learning methods for electric transportation”.

4)      In the section 2, Theoretical foundations and hypothesis delas with algorithm design but system description and related technical content is missing. Refer Investigation on parallel hybrid electric bicycle along with issuer management system for mountainous region.

5)      The contributions of this work are weak, the authors mention that five headway classifications, annual average daily traffic, and road gradient were taken as influential parameters but contribution of existing state-of-the-art protocols for classification and its findings is poorly framed. Similarly other objectives investigation and assessment is missing. Refer   assessment of electric two-wheeler ecosystem using novel pareto optimality and topsis methods for an ideal design solution.

6)       The objective is not clear and must map with methodology you have used to target the research objective.

7)      The conclusion is delivered with limited content. For this manuscript, the conclusion should be provided with more details, and add some quantitative research outcomes in the subsections.

8)      There is no coherence between each section titles, and the logic of the entire manuscript is not clear enough.

9)      Quality of figure 1 and 3 is poor need to enhance with higher accuracy percentage.

10)  The comparison of validation estimation outcomes using various error techniques is informative. However, the methodology lacks detailed information on the specific dataset used, the pre-processing steps, and the parameter settings for each algorithm. Providing these details would enhance the reproducibility and reliability of the results. The proposed methodology is too narrow and may be elaborated with comparison with similar work. Refer  Insight into the novel low cost green air pollution monitoring and control systems: A technological solution from concept to market.

Comments on the Quality of English Language

Moderate language polishing required 

Author Response

Reviewer 3:

Thank you for the review. We have tried to understand your comments and suggestions and fulfill them. Also, we found that some citations are not related to our paper and unfortunately are not appropriate to cite.

The authors present research on “A Novel DEA-IFRN SWARA-IFRN WASPAS model for efficiency of road sections based on headway analysis” is interesting work and study begin from the development of a novel interval fuzzy rough number decision-making model consisting of DEA, IFRN SWARA, and IFRN WASPAS methods. The contribution covers the study is a new extension of WASPAS method with interval fuzzy rough numbers. However, there are some questions remaining to be explained and the manuscript should be updated according to the follows.

------------------------------------------------------------------------------------------------------------

Following observation for improvement:

Comment 1: The abstract should be structured with background of WASPAS model and verification based on fuzzy model followed by different evaluation methods, key factors that influence accuracy in these different schemes and briefly explain key findings and conclusion for better clarity. Refer Electric vehicle battery capacity degradation and health estimation using machine-learning techniques: a review.

Reply: Thank you for your comment. We have modified the abstract:

The capacity of transport infrastructure is one of very important tasks in transport engineering, depending mostly on the geometric characteristics of road and headway analysis. In this paper, we have considered 14 road sections and determined their efficiency based on headway analysis. We have developed a novel interval fuzzy rough number decision-making model consisting of DEA (Data envelopment analysis), IFRN SWARA (Interval-valued fuzzy-rough number Stepwise Weight Assessment Ratio Analysis), and IFRN WASPAS (Interval-valued fuzzy-rough number Weighted Aggregates Sum Product Assessment) methods. The main contribution of this study is a new extension of WASPAS method with interval fuzzy rough numbers. Firstly, the DEA model has been applied to determine the efficiency of 14 road sections according to seven input-output parameters. Seven out of 14 alternatives show full efficiency and have been implemented further in the model. After that, the IFRN SWARA method has been used for the calculation of final weights, while IFRN WASPAS has been applied for ranking seven road sections. Results show that two sections are very similar and have almost equal efficiency, while other results are very stable. According to the results obtained, the best-ranked is a measuring segment of the Ivanjska-Šargovac section with a road gradient=-5.5%, which has low deviating values of headways according to the measurement classes from PC-PC to AT-PC, which shows balanced and continuous traffic flows. Finally, verification tests such as changing criteria weights, comparative analysis, changing parameter λ, and reverse rank analysis have been performed.

Comment 2: In the introduction section, different input and system operating conditions of fuzzy model schemes and brand concepts on leading vehicle and on its vehicle-dynamic characteristic and verification schemes are not clearly stated. Refer analyzing electric vehicle battery health performance using supervised machine learning.

Reply: The introduction section has been changed, while references have been cited in conclusion as part of future research.

Comment 3:  In the introduction section objective and methodology must be added for better understanding, reproducibility. Look at “a comprehensive review of categorization and perspectives on state-of-charge estimation using deep learning methods for electric transportation”.

Reply: The contributions and novelty of the paper are reflected in the following facts.

  • In accordance with the previously defined importance of headways for the entire area of road transport, a total of 14 sections of road infrastructure were consid-ered and a new model to determine their efficiency was created. In addition to five headway classifications, AADT (Annual average daily traffic) and road gra-dient were taken as influential parameters.
  • A new multiphase efficiency model, which includes the DEA model, SWARA and WASPAS methods in the form of interval fuzzy rough numbers, has been created. IFRNs have been used due to their ability to treat uncertainty in the deci-sion-making process adequately. The greatest contribution of the paper can be seen from the aspect of a new algorithm of the IFRN WASPAS method, which, according to the authors' knowledge, is presented for the first time in the litera-ture. So far, certain comparative analyses have been presented, but without the algorithm of this method.
  • Another aspect of contribution is reflected through the sustainable management of road infrastructure based on the results obtained and future recommendations. On the practical aspect, the study provides valuable insights for infrastructure managers and traffic experts, helping them make informed decisions to optimize road section efficiency.

Research gaps can be described in the following. The paper thoroughly analyzes road section efficiency by considering multiple input-output parameters, determining crite-ria weights by applying the IFRN SWARA method, sorting road sections with the IFRN WASPAS method, and performing verification tests. This comprehensive model en-hances the credibility and reliability of the research findings.

Comment 4:   In the section 2, Theoretical foundations and hypothesis delas with algorithm design but system description and related technical content is missing. Refer Investigation on parallel hybrid electric bicycle along with issuer management system for mountainous region.

Reply: What hypothesis is in the second section? The second section elaborates on previous studies, with no hypothesis. Your comment isn’t appropriate.

Comment 5: The contributions of this work are weak, the authors mention that five headway classifications, annual average daily traffic, and road gradient were taken as influential parameters but contribution of existing state-of-the-art protocols for classification and its findings is poorly framed. Similarly other objectives investigation and assessment is missing. Refer   assessment of electric two-wheeler ecosystem using novel pareto optimality and topsis methods for an ideal design solution.

Reply: Contributions are well elaborated in introduction and conclusion sections, while reference has been cited as part of future research.

Comment 6: The objective is not clear and must map with methodology you have used to target the research objective.

Reply: Thank you for your suggestion. Please read introduction and conclusion.

Comment 7: The conclusion is delivered with limited content. For this manuscript, the conclusion should be provided with more details, and add some quantitative research outcomes in the subsections.

Reply: Thank you for your positive comment. Conclusion is as follows:

Based on the development of the model and its application in a specific case study on 14 initial measuring segments of the given sections, ranking from the aspect of the efficiency of traffic flows on the sections was conducted. By generalizing and modeling the applied criteria of AADT, gradients and AM headways, it was shown that the rank of efficiency is achieved with a lower deviation of headways according to the specified vehicle classes. Also, the measure of efficiency implies an optimal flow, balanced and continuous, which is especially noticeable for the measuring segment of the Ivanjska-Šargovac section on a descent of - 5.5% where there is a low headway devia-tion. Too high headway deviations imply an unbalanced flow, and are often noticeable on an ascent, which caused some of those sections to be ranked significantly worse in terms of efficiency measures. Using a specific real case study that contains 14 road sec-tions gives practical relevance to the research. Analyzing real data and using the pro-posed model to evaluate efficiency based on headway analysis, the research provides actionable insights for improving traffic flow and infrastructure management.

In this paper, a novel integrated DEA-IFRN SWARA-IFRN WASPAS model has been developed to determine the efficiency of 14 road infrastructure sections. The con-tribution of the research can be viewed in two ways, from a scientific-methodological and professional aspect. From the scientific aspect, it is certainly a dominant contribu-tion, which is reflected in the development of the IFRN WASPAS method, while from the professional aspect, it represents support for the infrastructure manager and traffic experts in order to define certain measures. Also, the proposed innovative model has the potential to advance the field of transportation engineering by providing a more comprehensive analysis of infrastructure efficiency.

On the other hand, the findings of the study may have limited generalizability due to the focus on a specific set of road sections and the relatively small sample size. This restricts the broader applicability of the developed model to different geographical contexts or transportation systems, potentially limiting its usefulness to practitioners in diverse settings. Limitations related to this research can be manifested through rela-tively short measuring segments (with a length of 1000 meters) and a small number of DMs who participated in group decision-making. Also, one of the limitations may be the lack of new data related to AADT or the fact that integration of multiple deci-sion-making methods and the use of interval fuzzy rough numbers may introduce methodological complexity, making it challenging for readers to grasp the intricacies of the approach. This complexity could hinder understanding and replication by other researchers or practitioners, potentially limiting the adoption of the developed model. These limitations can be mitigated if the reproduction of the model will be made soon with more parameters and if the model will be applied under the advice of experts in the field of methodology.

Future research refers to the collection of data and the determination of the effi-ciency of new sections of road infrastructure, as well as the definition of additional parameters and the inclusion of DMs of different structures. The developed model can be applied in any other case study that contains multiple variants and criteria. Also, from a methodological aspect future research can be related to extension methods in other forms like quasirung fuzzy sets [43], polytopic fuzzy sets [44], integration with machine learning [45,46], multi-objective optimization [47] etc.

Comment 8: There is no coherence between each section titles, and the logic of the entire manuscript is not clear enough.

Reply: Some part of the paper have been changed and polished from this aspect.

Comment 9: Quality of figure 1 and 3 is poor need to enhance with higher accuracy percentage.

Reply: Figures have been replaced.

Figure 1. New criteria weights in 70 cases

Figure 3. Ranks in CA

Comment 10: The comparison of validation estimation outcomes using various error techniques is informative. However, the methodology lacks detailed information on the specific dataset used, the pre-processing steps, and the parameter settings for each algorithm. Providing these details would enhance the reproducibility and reliability of the results. The proposed methodology is too narrow and may be elaborated with comparison with similar work. Refer  Insight into the novel low cost green air pollution monitoring and control systems: A technological solution from concept to market.

Reply: Please read the fourth section which provides all necessary information. Also, unfortunately, the proposed article isn’t appropriate for citation in our paper because hasn't a clear relationship.

Collection and processing of data

In order to determine road section efficiency based on headways, data was collected for a total of 14 sections with the following characteristics. The length of a measuring segment on which the measurements were made is at least 1000 meters long, the road gradient ranges from -5.50 to 7.50%, the section length is from 7.45 km to 38.55 km, while the size of the measurement sample varies from 713 to 1011. Also, it is important to note that headway was measured for all types of vehicles. The characteristics of the measuring sections are presented in Table 1.

Table 1. Characteristics of road sections and average headways

Technical and operational characteristics of the sections

Measurement sample size (no. of measurements)

AM of headways by measuring segments of sections

Section symbol

Measuring segment length

Road gradient

  Section length (km)

Th [s]       (PC-PC)

Th [s]       (LDV-PC)

Th [s]       (HDV-PC)

Th [s]       (BUS-PC)

Th [s]       (AT-PC)

DMU1

M-I-108

min 1000 m

-5.50%

14.967

1000

5.349

7.714

7.577

8.633

10.136

DMU2

M-I-103

min 1000 m

-5.00%

14.073

1000

19.269

25.272

40.269

29.491

44.767

DMU3

M-I-108

min 1000 m

-3.00%

14.967

1000

5.331

9.406

9.107

4.979

7.934

DMU4

M-I-108

min 1000 m

1.50%

14.967

1000

12.553

21.713

19.393

23.681

12.926

DMU5

M-I-103

min 1000 m

-1.00%

14.073

1010

18.174

21.102

36.395

26.04

33.222

DMU6

M-I-105

min 1000 m

0%

7.405

1011

3.6

8.609

11.969

10.162

16.488

DMU7

M-I-106

min 1000 m

1.00%

38.553

912

9.794

25.096

18.537

26.949

16.669

DMU8

M-I-108

min 1000 m

2.00%

16.734

775

24.22

65.23

66.46

64.88

97.99

DMU9

M-I-106

min 1000 m

3.00%

38.553

908

12.43

31.124

24.458

21.191

64.944

DMU10

M-I-106

min 1000 m

4.00%

38.553

1007

8.48

31.271

31.253

34.794

63.897

DMU11

M-I-103

min 1000 m

5.00%

14.073

918

4.79

32.367

36.991

36.362

82.858

DMU12

M-I-108

min 1000 m

6.00%

20.134

736

12.907

82.888

126.711

118.67

189.593

DMU13

M-I-108

min 1000 m

7.00%

20.134

713

14.739

90.027

132.791

135.949

236.574

DMU14

M-I-108

min 1000 m

7.50%

20.134

811

16.559

97.84

141.196

146.706

264.434

DMU (Decision making unit), PC (Passenger car), LDV (Light-Duty Vehicle), HDV (Heavy-Duty Vehicle), AT (Auto train)

After collection, processing and sorting of data, a DEA model [34] has been applied based on a total of seven parameters, which later represent criteria in the MCDM model. In addition to the given parameters related to the headway and road gradient, AADT is included as an additional parameter. C1 – AADT, C2 – road gradient, C3 - Th [s] (PC-PC), C4 - Th [s] (LDV-PC), C5 - Th [s] (HDV-PC), C6 -Th [s] (BUS-PC), C7 - Th [s] (AT-PC).

Reviewer 4 Report

Comments and Suggestions for Authors

The manuscript is completely clear. The way and style of presentation are very clear. All other sections of the article are consistent and balanced. The goals and objectives of the study are clearly formulated, the results are correctly interpreted.  The conclusions need improvement. The calculation results are presented in the form of figures and tables that are easy to understand. 

Author Response

Reviewer 4:

Thank you very much for your positive review.

The manuscript is completely clear. The way and style of presentation are very clear. All other sections of the article are consistent and balanced. The goals and objectives of the study are clearly formulated, the results are correctly interpreted.  The conclusions need improvement. The calculation results are presented in the form of figures and tables that are easy to understand. 

 ------------------------------------------------------------------------------------------------------------

Based on the development of the model and its application in a specific case study on 14 initial measuring segments of the given sections, ranking from the aspect of the efficiency of traffic flows on the sections was conducted. By generalizing and modeling the applied criteria of AADT, gradients and AM headways, it was shown that the rank of efficiency is achieved with a lower deviation of headways according to the specified vehicle classes. Also, the measure of efficiency implies an optimal flow, balanced and continuous, which is especially noticeable for the measuring segment of the Ivanjska-Šargovac section on a descent of - 5.5% where there is a low headway deviation. Too high headway deviations imply an unbalanced flow, and are often noticeable on an ascent, which caused some of those sections to be ranked significantly worse in terms of efficiency measures. Using a specific real case study that contains 14 road sections gives practical relevance to the research. Analyzing real data and using the proposed model to evaluate efficiency based on headway analysis, the research provides actionable insights for improving traffic flow and infrastructure management.

In this paper, a novel integrated DEA-IFRN SWARA-IFRN WASPAS model has been developed to determine the efficiency of 14 road infrastructure sections. The contribution of the research can be viewed in two ways, from a scientific-methodological and professional aspect. From the scientific aspect, it is certainly a dominant contribution, which is reflected in the development of the IFRN WASPAS method, while from the professional aspect, it represents support for the infrastructure manager and traffic experts in order to define certain measures. Also, the proposed innovative model has the potential to advance the field of transportation engineering by providing a more comprehensive analysis of infrastructure efficiency.

On the other hand, the findings of the study may have limited generalizability due to the focus on a specific set of road sections and the relatively small sample size. This restricts the broader applicability of the developed model to different geographical contexts or transportation systems, potentially limiting its usefulness to practitioners in diverse settings. Limitations related to this research can be manifested through relatively short measuring segments (with a length of 1000 meters) and a small number of DMs who participated in group decision-making. Also, one of the limitations may be the lack of new data related to AADT or the fact that integration of multiple decision-making methods and the use of interval fuzzy rough numbers may introduce methodological complexity, making it challenging for readers to grasp the intricacies of the approach. This complexity could hinder understanding and replication by other researchers or practitioners, potentially limiting the adoption of the developed model. These limitations can be mitigated if the reproduction of the model will be made soon with more parameters and if the model will be applied under the advice of experts in the field of methodology.

Future research refers to the collection of data and the determination of the efficiency of new sections of road infrastructure, as well as the definition of additional parameters and the inclusion of DMs of different structures. The developed model can be applied in any other case study that contains multiple variants and criteria. Also, from a methodological aspect future research can be related to extension methods in other forms like quasirung fuzzy sets [43], polytopic fuzzy sets [44], integration with machine learning [45,46], multi-objective optimization [47] etc.

Round 2

Reviewer 1 Report

Comments and Suggestions for Authors

The authors have substantially improved the quality of the manuscript by considering all the comments and suggestions of the Reviewer. The revised version may be accepted in its present form.

Reviewer 2 Report

Comments and Suggestions for Authors

The majority of our remarks have been addressed and revised. Consequently, I recommend accepting the paper.